# An observationally-constrained estimate of global dust aerosol optical depth

David A. Ridley[1], Colette L. Heald[1], Jasper F. Kok[2], Chun Zhao[3]

[1]Civil and Environmental Engineering, Massachusetts Institute of Technology
5   [2]Department of Atmospheric and Oceanic Sciences, UCLA
[3]Atmospheric Sciences and Global Change Division, Pacific Northwest National Lab, Richland, WA 99352

*Correspondence to:* David A. Ridley (daridley@mit.edu)

**Abstract.** The role of mineral dust in climate and ecosystems has been largely quantified using global climate and chemistry model simulations of dust emission, transport, and deposition. However, differences between these model simulations are 10 substantial, with estimates of global dust aerosol optical depth (AOD) that vary by over a factor of 5. Here we develop an observationally-based estimate of the global dust AOD, using multiple satellite platforms, in-situ AOD observations and four state-of-the-science global models over 2004 - 2008. We estimate that the global dust AOD at 550 nm is $0.030 \pm 0.005$ (1$\sigma$), higher than the AeroCom model median (0.023) and substantially narrowing the uncertainty. The methodology used provides regional, seasonal dust AOD and the associated statistical uncertainty for key dust regions around the globe with which model 15 dust schemes can be evaluated. Exploring the regional and seasonal differences in dust AOD between our observationally-based estimate and the four models in this study, we find that emissions in Africa are often overrepresented at the expense of Asian and Middle-Eastern emissions, and that dust removal appears to be too rapid in most models.

## 1 Introduction

Mineral dust is a key component of aerosol, affecting climate through interaction with radiation, clouds and snowpack, 20 human health through contribution to particulate matter (PM), and ecosystem health through nutrient transport and deposition. The direct radiative effect (DRE) of dust contributes ~30% of the total aerosol global mean DRE (Heald et al., 2014); however, there is significant uncertainty in the radiative forcing of dust, estimated to be anywhere between -0.3 and +0.1 Wm$^{-2}$ (Boucher et al., 2013), owing to large uncertainties in the anthropogenically-driven changes in dust (Ginoux et al., 2012; Heald and Spracklen, 2015), the particle morphology and absorption properties (e.g. Balkanski et al., 2007; Mishra et al., 2008), and the 25 dust size distribution (Kok, 2011; Kok et al., in review). Dust concentrations are often highest in remote regions that are sparsely-monitored, leading to further uncertainty on the atmospheric burden and the associated radiative effects.

Dust aerosol can be transported far downwind of desert source regions, having a significant impact on the surface PM thousands of kilometers downwind (Prospero, 2007; Prospero et al., 2014; Zhang et al., 2013). This poses a significant health concern through cases of premature mortality from respiratory and cardiovascular disease that are attributed to aerosol 30 exposure (Lim et al., 2012). Studies attempting to quantify the global premature mortality from aerosol exposure (e.g. van

Donkelaar et al., 2006; Evans et al., 2013) highlight the strong contribution of dust to PM across large regions of Africa, Asia and the Middle East. Because of the lack of surface monitoring in dust influenced regions, those studies rely on satellite observations of aerosol optical depth (AOD), a measure of the column-integrated aerosol that is critical for understanding the radiative effect. Relating the AOD to surface PM requires information on the vertical distribution and aerosol speciation, generally obtained from models, which can introduce considerable uncertainty (Ford and Heald, 2015). Limited observations of global dust aerosol hinder our ability to estimate the full extent of the climate and air quality impacts of mineral dust.

To simulate the global dust cycle, models must be able to predict the vertical dust flux from suitable regions and represent the evolution of the particle size distribution while the dust is transported and deposited out of the atmosphere (e.g. Kok et al., 2012). The AeroCom project, an intercomparison and evaluation of different aerosol models, provides a detailed evaluation of dust aerosol simulations from multiple models (Huneeus et al., 2011). There is a considerable spread in global dust AOD estimates from models ranging from 0.010 to 0.053 (yielding a mean of 0.028 ± 0.011) and an AeroCom "median model" estimate of 0.023. The uncertainty in the AOD highlights the underlying uncertainties in emissions, size distributions, lifetime, and optical properties. Even over the well-studied, most productive dust region of West Africa, climate models struggle to represent dust emission and their year-to-year changes (Evan et al., 2014).

An observationally-constrained estimate of dust AOD can thus provide a valuable metric to holistically evaluate model dust emission, transport, and deposition, thereby helping constrain both the direct radiative effect and the role of dust in adverse health effects from exposure to PM. Here we derive such a metric, with consideration for the sources of uncertainty, and use it to highlight seasons and regions in which current global models deviate from the observations.

## 2 Data Description

To derive an estimate of dust AOD we make use of AOD retrievals from three satellite instruments as well as surface-based sun photometers to provide a 'ground-truth' for correcting the satellite retrievals. We use four global aerosol models to provide a range of estimates for the non-dust aerosol AOD and the spatial distribution of dust aerosol (see Section 3 for a full description of the methodology). We use observational data and model simulations over the 5-year period between 2004 and 2008, except when calculating biases between satellite and surface-based observations, for which we leverage a longer dataset between 2003 and 2013. Below we give a brief description of each instrument and model, and the products used.

### 2.1 Moderate Resolution Imaging Spectroradiometer (MODIS)

Two MODIS instruments are in sun-synchronous orbit aboard the Terra and Aqua platforms, making equatorial overpasses at 10.30am and 2.30pm local time (LT), respectively. Radiance measurements are made across 36 bands between 0.4 and 14 microns, with 7 channels used to retrieve the AOD at 550 nm. The wide swath (2330 km) allows almost daily coverage of the globe by both instruments at a native resolution of 500 m at nadir (2 km at swath edge), for the aerosol-relevant bands, with AOD reported at approximately 10 km x 10 km resolution (Level-2 product). The Collection 6 MODIS data

includes a merged AOD product that combines retrievals over ocean, vegetated land surface (Dark Target), and bright land surface (Deep Blue) to maximize global coverage. The retrieved AOD ($\tau$) is estimated to be accurate to $\pm 0.03 \pm 0.05\tau$ over ocean (Remer et al., 2005), $\pm 0.05 \pm 0.15\tau$ over dark land surfaces (Levy et al., 2010) and $\pm 0.05 \pm 0.20\tau$ over bright surfaces (Hsu et al., 2006; Sayer et al., 2013). The quality-assured (QA) Level-2 AOD retrievals are aggregated on a daily basis onto a

$1° \times 1°$ grid (Level-3) with statistics, including cloud fraction and standard deviation. Throughout this study we use the Level-3 product. The merged Level-3 product uses QA=3 data over land and QA=1-3 data over ocean, where higher quality data is given commensurate weighting. Baddock et al. (2016) show that correlation between the frequency of high AOD and dust source location is actually improved when using only QA=1 data. For data to be considered QA>1 the standard deviation in AOD between 1km retrievals must remain below a threshold of 0.18. Therefore, some legitimate dust-influenced retrievals

over source may be discarded when using the Level-3 merged product. However, this is a trade off in terms of improving the quality of the retrieval away from source regions. The MODIS retrieval algorithm uses a look-up table of surface reflectance for a set of simulated aerosol properties to determine the AOD that best represents the observed reflectance. For the Deep Blue retrieval, the most relevant to this study over dust-influenced regions, the assumed optical properties of the dust aerosol have a single-scattering albedo (SSA) between 0.87 and 1.0 for the look-up tables at 412 nm and 490 nm and a refractive index of

$1.55 – 0.0i$ (at 670 nm). The Mie calculation uses an effective phase function, derived from comparison of the Sea-Viewing Wide Field-of-View Sensor (SeaWIFS) instrument retrievals with AERONET, over the ocean to account for non-sphericity. Different locations and loading conditions trigger changes in the wavelengths used in the retrieval, more information can be found in Hsu et al. (2004, 2013)

## 2.2 Multi-angle Imaging Spectro-Radiometer (MISR)

The MISR instrument, aboard the Terra satellite platform, measures radiance over 9 camera angles with an equatorial overpass at 10.30am LT. The relatively narrow swath width (400 km) results in global coverage every 9 days, compared with 1 - 2 days by MODIS. MISR provides AOD at four wavelengths (446nm, 558nm, 672nm, 867nm) with about three-quarters of retrievals falling within $0.20\tau$ (but no less than 0.05) of AERONET observations (we assume an instrument uncertainty of $\pm 0.05 \pm 0.20\tau$ throughout this study) and reliable retrieval over bright desert surfaces (Kahn et al., 2010; Martonchik et al.,

1998, 2004). In this study, we use the Level-3 daily $0.5° \times 0.5°$ resolution gridded AOD product. The MISR retrieval algorithm uses simulated TOA radiances using properties for eight particle types to determine the AOD. The optical properties of the two aerosol particle types corresponding to dust assume a refractive index of $1.51 – 6.5\times10^{-4}i$ and SSA between 0.971 and 0.994 (at 672 nm). The extinction is calculated using the discrete dipole approximation and the T-matrix technique to account for particle non-sphericity (Kalashnikova et al., 2005; Martonchik et al., 2009).

**2.3 Aerosol Robotic Network (AERONET)**

AERONET consists of a global network of Cimel Electronique CE-318 sun photometers, which reports AOD with a high degree of accuracy leading to estimated errors of ~0.01 - 0.02 (Eck et al., 1999; Holben et al., 1998). Direct sun measurements

are made every 15 minutes at 340, 380, 440, 500, 675, 870, 940 and 1020 nm and AOD is retrieved at all but the 940 nm channel, which is used to provide total column water vapor. We use Level 2.0 data that has been screened for clouds (Smirnov et al., 2000). The wavelength-dependence of the AOD, described by the angstrom exponent (Ångström, 1964) between the AOD at 440 and at 870 nm, is used to distinguish AOD dominated by coarse aerosol that is indicated by a lower angstrom

exponent than for fine aerosol (e.g. O'Neill et al., 2001; Reid et al., 1999). Sun photometer measurements made from aboard ship cruises as part of the AERONET Marine Aerosol Network (MAN; Smirnov et al., 2011) are incorporated into the AERONET analysis in this work.

## 2.4 GEOS-Chem

We use the GEOS-Chem global chemical transport model (v9-01-01; http://www.geos-chem.org) to simulate the coupled

oxidant-aerosol chemistry of the troposphere at a resolution of 2.5° by 2.0° over 47 vertical levels following the specifications used in (Heald et al., 2014). The oxidant-aerosol simulation includes $H_2SO_4$-$HNO_3$-$NH_3$ aerosol thermodynamics described by ISORROPIA II (Fountoukis and Nenes, 2007) and coupled with an $O_3$-$NO_x$-hydrocarbon chemical mechanism (Park et al., 2004, 2006). The aerosol simulation also includes carbonaceous aerosols (Park et al., 2003; Pye et al., 2010; Pye and Seinfeld, 2010), mineral dust (Fairlie et al., 2007; Ridley et al., 2012), and sea salt (Alexander et al., 2005). Aerosol mass is transported

in 4 size bins (0.1–1.0, 1.0–1.8, 1.8–3.0, and 3.0–6.0 µm radius) for dust, two for sea-salt and one for each of the other species. The model is driven by assimilated meteorology from the NASA Modern-Era Retrospective analysis for Research and Applications (MERRA), which provides winds, precipitation, could cover etc. at 1-hourly and 3-hourly temporal resolution. Dust emissions are generated using the DEAD scheme (Zender, 2003) with the GOCART source function (Ginoux et al., 2001; Prospero et al., 2002) and a fixed soil clay fraction of 0.2. We follow Ridley et al. (2013) by using a probability distribution of

sub-grid scale winds, generated from 0.5° x 0.67° MERRA 10-m winds, rather than the average wind speed when calculating dust uplift. Biomass burning emissions are provided by the Global Fire Emissions Database version 3 (GFEDv3; van der Werf et al., 2010). Anthropogenic emissions are provided by the Emissions Database for Global Atmospheric Research (EDGAR) v3.2 inventory (Olivier, 2001) for $SO_x$, $NO_x$, and CO which is superseded by the National Emissions Inventory (NEI99; http://www.epa.gov/ttn/chief/net/1999inventory.html) over the United States and Streets et al. (2003, 2006) over Asia (van

Donkelaar et al., 2008). Sea salt emissions follow Gong (2003) with added dependence on sea surface temperature (Jaeglé et al., 2011). AOD at 550nm is calculated online assuming lognormal size distributions of externally mixed aerosols and is a function of the local relative humidity to account for hygroscopic growth (Martin et al., 2003). Aerosol optical properties are based on the Global Aerosol Data Set (GADS) (Hess et al., 1998a) with modifications to the size distribution based on field observations (Drury et al., 2010; Jaeglé et al., 2011) and improvements to the UV/visible refractive indices of dust (Sinyuk et

al., 2003).

## 2.5 CESM

The Community Earth System Model (CESM), version 1.1 (Hurrell, 2013), is used in this study following the specifications described in Kok et al. (2014b). The atmospheric component of the model, the Community Atmospheric Model version 4 (CAM4), is run at 2.5° x 1.9° resolution and is driven by ERA-Interim reanalysis meteorology (Dee et al., 2011)
with free-running dynamics. CAM4 simulates aerosol as bulk species from the Model for OZone And Related chemical Tracers (MOZART) chemistry package (Lamarque et al., 2012), including sulfate, ammonium, ammonium nitrate, black carbon, organic carbon and secondary organic aerosol. Emissions of these species are prescribed by the AeroCom specifications (Neale et al., 2010). Sea salt is emitted and transported in four size bins and is calculated from 10 m wind speed (Mahowald et al., 2006). Dust emission in the Community Land Module version 4 (CLM4) is traditionally based on the DEAD
dust scheme (Zender, 2003) with some minor modifications (Mahowald et al., 2006, 2010). Here we use the new dust emission model developed in Kok et al. (2014a), which generates a vertical dust flux with no prescribed source function, and accounts for the exponential increase in dust flux with increasing soil erodibility. This dust emission model both better reproduces small-scale dust emission measurements (Kok et al., 2014a) and its implementation in CESM results in improved agreement against AERONET measurements in dusty regions (Kok et al., 2014b).. Dust is emitted into four size bins (0.1–1.0, 1.0–2.5, 2.5–5.0,
and 5.0–10 μm diameter), and the fraction emitted into each bin is independent of wind speed, as shown by measurements (Kok, 2011b), and distributed following brittle fragmentation theory (Kok, 2011a). All aerosols are assumed externally mixed. The aerosol optical properties are based on GADS (Hess et al., 1998a) with improvements to the dust optical properties described in Albani et al. (2014) And prescribed size distributions that can be found in (Emmons et al., 2010).

## 2.6 WRF-Chem

The quasi-global configuration of the WRF-Chem (version 3.5.1) model is used in this study, described in detail in Hu et al. (2016). The simulation uses the MOSAIC (Model for Simulation Aerosol Interactions and Chemistry) aerosol module (Zaveri et al., 2008) with the CBM-Z (carbon bond mechanism) photochemical mechanism (Zaveri and Peters, 1999). A sectional approach is used to represent aerosol size distributions with eight discrete size bins and all major aerosol components including sulfate ($SO_4^{-2}$), nitrate ($NO_3^-$), ammonium ($NH_4^+$), black carbon (BC), organic matter (OM), sea-salt,
methanesulfonic acid, and mineral dust are simulated. The MOSAIC aerosol scheme includes physical and chemical processes of nucleation, condensation, coagulation, aqueous phase chemistry, and water uptake by aerosols. The model is run at a resolution of 1° x 1° (between 180° W-180° E and 67.5° S-77.5° N) with 35 vertical layers up to 50hPa (Hu et al., 2016). The modeled u and v wind components and temperature in the free atmosphere above the planetary boundary layer are nudged towards NCEP/FNL reanalyses on 6-hourly time steps (Stauffer and Seaman, 1990). Biomass burning emissions are derived
from GFEDv3. Anthropogenic emissions are provided by the REanalysis of the TROpospheric (RETRO) chemical composition inventories (http://retro.enes.org/index.shtml) except over East Asia, where emissions are taken from the inventory developed for the INTEX-B mission in 2006 (Zhang et al., 2009), updated with $SO_2$ and carbonaceous emissions

from Lu et al. (2011), and the United States, where the National Emissions Inventory (NEI) for 2005 are used. Sea salt emissions are based on Gong, (2003), with added emission dependence on sea surface temperature (Jaeglé et al., 2011). Dust emission fluxes are calculated with the GOCART dust emissions scheme (Ginoux et al., 2001) and partitioned into the MOSAIC size bins based on brittle fragment theory (Kok, 2011a). Aerosol optical properties are computed as a function of

wavelength for each model grid box. Aerosols are assumed internally mixed (a volumetric mean refractive index) in each bin. The Optical Properties of Aerosols and Clouds (OPAC) dataset (Hess et al., 1998b) is used for the shortwave (SW) and longwave (LW) refractive indices of aerosols, except that a constant value of $1.53 + 0.003i$ is used for the SW refractive index of dust following Zhao et al. (2010, 2011). A detailed description of the aerosol optical properties calculated in WRF-Chem can be found in Fast et al. (2006) and Barnard et al. (2010). The optical properties and direct radiative forcing of individual

aerosol species in the atmosphere are diagnosed following the methodology described in Zhao et al. (2013).

### 2.7 MERRAero

The NASA Global Modeling and Assimilation Office (GMAO) GEOS-5 Earth system model can be run in a configuration that assimilates meteorological and aerosol properties retrieved from NASA Earth observing satellite platforms (Rienecker et al., 2011). The resulting aerosol simulation is termed MERRAero. The simulation is run at a resolution of 0.5°

x 0.625° providing speciated AOD with 3-hourly temporal resolution. The aerosol processes are based on the Goddard Chemistry, Aerosol, Radiation and Transport model (GOCART; Chin et al., 2002) with coupling of chemistry and climate (Colarco et al., 2010). Dust, sulfate, organic carbon, black carbon and sea salt are simulated as external mixtures. Dust aerosol is partitioned into 8 size bins between 0.1 and 10 µm particle radius, and sea salt aerosol is partitioned into 5 size bins between 0.03 and 10 µm dry radius; all other aerosol is transported in a single size bin per species. Emissions of fossil fuels and biofuel

follow the GOCART model (Chin et al., 2002) with updates in the U.S. following Park et al. (2003). $SO_2$ emissions are from the EDGAR-4.1 inventory with altered injection profiles (Buchard et al., 2014) and biomass burning emissions are supplied from the NASA Quick Fire Emission Dataset (QFED) Version 2.1. Sea salt aerosol production follows Gong (2003) with added dependence on sea surface temperature (Jaeglé et al., 2011). The aerosol optical properties follow GADS, but with modifications to reduce the absorption of dust at short wavelengths (Sinyuk et al., 2003), and extinction is calculated following

Mie theory assuming spherical particles (Colarco et al., 2010). MERRAero differs from the other three models used in that the model assimilates AOD information from the MODIS instruments. The assimilation process is explained in detail in Buchard et al. (2016), here we give a brief description. The MODIS reflectances are cloud screened and converted to AOD using a neural net framework. The error covariance between the 2D MODIS AOD and the model AOD is used to generate 3D aerosol mass increments. Using a Local Displacement Ensemble (LDE) methodology, ensembles of isotropic displacements in aerosol

mass around a central grid box are weighted based upon the reduction in the error. Different aerosol species can be perturbed in each vertical layer, e.g. to allow a plume to be shifted to better match the MODIS AOD, and therefore aerosol mass can vary independently for each species.

## 3 Methodology

### 3.1 Derivation of dust AOD

Our aim is to provide seasonal dust AOD estimates, both global and regional, that are as independent from modeled dust estimates as possible. The methodology and development of associated uncertainty estimates is described in detail below, but the general methodology is as follows: We rely primarily on satellite retrievals of AOD, which we bias-correct using dust-dominated AERONET AOD retrievals. In order to partition the retrieved AOD in dusty regions between the component due to dust and the component due to other aerosols, we use simulated estimates of non-dust AOD with several global models in 15 regions that are identified as contributing significantly to the global dust AOD. These regions are defined such that they account for the majority of dust AOD, based on model estimates, and are shown in Figure 1. Finally, model dust AOD is used to estimate the fraction of dust AOD that is outside of the 15 dust-dominated regions, thereby providing global seasonal dust AOD estimates between 2004 and 2008. Our methodology accounts for many uncertainties, including the satellite retrieval error, estimation of seasonal mean AOD, bias correction, modelled non-dust AOD and global scaling factors. We discuss potential biases that are not accounted for in Section 4.4.

We aggregate daily AOD data from MISR and both MODIS instruments (Aqua and Terra) onto a 2° x 2.5° grid and average over 3-month periods to increase coverage and provide a consistent grid between model and observations. We use bootstrapping (Efron and Gong, 1983) to estimate the random uncertainty in the seasonally-averaged AOD due to sampling uncertainty within each grid box. This is achieved by randomly sampling (with replacement) the grid box daily AOD $n$ times, where $n$ is the number of days with a retrieval in that 3-month period, and the mean calculated. This is repeated to build a probability distribution of the seasonal AOD for each grid box. We find that a log-normal distribution is a good approximation to the resulting seasonal AOD uncertainty distribution, so we retain the mean and standard deviation of this distribution as the mean and uncertainty on the seasonal $\log_{10}(AOD)$ for each grid box.

Although the bootstrapping method quantifies the random error in each grid box' seasonal AOD, it does not quantify or correct the systematic error (bias) in the AOD. Therefore, we use AERONET AOD as ground-truth to apply a bias correction to the satellite-retrieved AOD, with a focus on dust influenced regions. AERONET hourly AOD (interpolated to 550nm) is used to produce a morning (10am – noon, LT) and afternoon (1pm – 3pm, LT) average to compare with daily retrievals from aboard the Terra and Aqua satellites, respectively. We compare these at the native satellite retrieval resolution and choose a two-hour window to both cover the approximate range of the overpass times and to maximize the number of coincident AERONET and satellite AOD retrievals. We use all AERONET sites within the regions defined in Figure 1 and aggregate all data in each region for the comparison. For the regions encompassing ocean we also use AOD measurements from the AERONET Marine Aerosol Network (MAN) where available (see Figure 1). We generate histograms of the daily $\log_{10}(AOD)$ from AERONET and each satellite instrument using data between 2003 and 2013 (see Figures S1 – S3) and present the statistics of the bias and linear regression for each region in Table 1. Although 2004 – 2008 is the main period of study in this

research we use 12 years of AERONET and satellite retrievals to maximize the amount of data and better characterize the biases. More than 100 days of co-located data available are required for the bias correction to be applied to a region; a criterion usually met, except for MISR in some Asian deserts. The standard deviation of the bias correction is derived from each pair of daily $\log_{10}(AOD)$ in a region, and is used to propagate this uncertainty into the global dust AOD estimate (see below). If not enough data is available, we apply a bias correction of 1.0 (i.e. no bias) with a standard deviation of ±50% to represent the uncertainty.

On average across all sites, both MODIS instruments show a slight high mean annual bias in AOD relative to AERONET retrievals (+8% for Terra; +4% for Aqua; see Table 1), although there is considerable variability between regions and on a day-to-day basis (indicated by the correlation coefficient, $r$, in Table 1). MISR is also biased high relative to AERONET (+13%), owing primarily to retrievals when AOD<0.5, but also exhibiting a low bias for AOD>1.0, consistent with previous comparisons, e.g. Moon et al. (2015). Splitting the data by season does not yield qualitatively different results; however, it reduces the number of data points in some regions enough to make comparison unreliable. Therefore, we apply an annual bias correction per region. The bias correction has a moderate impact on the average global dust AOD, decreasing it by 10% and bringing the individual satellite instruments into closer agreement. However, the large uncertainty on the bias correction (see Table 1) is a major source of the uncertainty on the global dust AOD (see Table 2).

Although dust aerosol is often the main contributor to the AOD in the regions shown in Figure 1, other aerosol species can make a significant contribution and need to be accounted for to extract dust AOD from the satellite retrievals of AOD. We use GEOS-Chem, CESM, WRF-Chem and MERRAero to provide non-dust AOD; using multiple models provides an estimate of the variability in the non-dust portion of the AOD resulting from uncertainty in aerosol emissions and formation mechanisms. Anthropogenic aerosol is generally well characterized by global models, especially on seasonal timescales, and have been regularly evaluated against observations, particularly in the Northern Hemisphere (e.g. Hu et al., 2016; Leibensperger et al., 2012; Liu et al., 2012; Mann et al., 2014). We focus on regions in which the dust AOD often dominates to reduce potential errors from biases in modeled non-dust AOD. Biomass burning aerosol concentrations are inherently uncertain because of the challenges in determining burned area and emissions factors (French et al., 2004; van der Werf et al., 2006). Despite considerable evaluation against observations the resulting biomass burning AOD is sometimes underrepresented (Matichuk et al., 2007; Reddington et al., 2016); therefore, we treat regions affected by biomass burning emissions with caution. In the regions analysed, dust aerosol plays a key role and often dominates in the spring and summer, limiting the influence of the model non-dust AOD. Exceptions to this are in South America, South Africa, and Australia, that have a minimal impact on the global dust AOD, and the Gulf of Guinea, where significant biomass burning aerosol is present (we consider results with and without these regions, see Table 1). In addition, most regions considered in this study are inland and therefore sea salt aerosol will have a limited impact. Figure 2 displays the climatology of non-dust AOD and dust AOD for each model used, averaged over 2004 - 2008.

For each 2° x 2.5° grid box, $i$, within the 15 regions we apply the AERONET-derived bias correction, $\alpha$, to the seasonal satellite AOD, $\tau^{obs}$, and subtract the model non-dust AOD, $\tau^{model}_{nd}$, to provide an estimate of the regional dust AOD, $\tau^{reg}_d$ (Eqn. 1). We allow negative values of $\tau^{reg}_d$ so as not to introduce a positive bias. The uncertainty distribution for each of these three variables, bias correction, satellite $\log_{10}(AOD)$, and model non-dust $\log_{10}(AOD)$, is sampled and the average dust AOD is calculated for each region. This process is repeated multiple times to yield a stable distribution of seasonal dust AOD (200 times is sufficient for a robust average) for each of the regions between 2004 and 2008. For a single iteration of the dust AOD calculation we use the same random sampling (sampling the same number of sigma from the mean) for all grid boxes, thereby assuming the worst case scenario that the uncertainty is correlated spatially. If we use a different sampling of the uncertainty distribution for each grid box, the uncertainty on the global dust AOD drops by approximately a factor of 8.

$$\tau^{reg}_d = \frac{1}{N}\sum_i^N \alpha_i \tau^{obs}_i - \tau^{model}_{nd,i}$$
(1)

The regional dust AOD, $\tau^{reg}_d$, for the 15 regions is weighted by surface area, $A^{reg}$, summed, and scaled by the surface area of the Earth, $A_E$, to give the total regional contribution to the global dust AOD (Eqn. 2). To obtain the globally-averaged dust AOD, $\tau^{glob}_d$, we calculate the ratio, $\beta$, between the modeled dust AOD across all regions and the modeled global dust AOD (Eqn. 3).

$$\tau^{glob}_d = \beta \frac{1}{A_E} \Sigma_r^{N^{reg}} A^{reg} \tau^{reg}_d,$$
(2)

$$\beta = \frac{\tau^{glob,model}_d}{\frac{1}{A_E} \Sigma_r^{N^{reg}} A^{reg} \tau^{reg,model}_d}$$
(3)

This allows the satellite estimate within the regions to be scaled to a global dust AOD estimate. This is the only element of our analysis that relies upon simulated dust AOD. The 15 regions account for between 83% and 95% of the global dust AOD, depending on the model, so the model influence is limited and using multiple models provides an estimate of the uncertainty this introduces into our analysis (see Table 1). This process is repeated for all combinations of the 3 satellite instruments, 4 model estimates for non-dust, and 4 model regional-to-global scaling factors; this produces 48 realizations, 16 per satellite instrument, each with an uncertainty estimate. We use the kernel density estimation method (Silverman, 1986) with a Gaussian kernel and standard smoothing to determine a probability density function for the global dust AOD based on the 48 realizations.

## 4 Results

### 4.1 The observationally-constrained global dust AOD

The global dust AOD for the AeroCom models in Huneeus et al. (2011) is also displayed in Figure 3 with the associated probability density function generated using the kernel density estimation method. Our observational estimate of the global dust AOD is 0.030 ± 0.005 (1σ), and is thus much more narrowly constrained than the AeroCom estimate of 0.028 ± 0.011.

Over three-quarters (77%) of the ensemble members fall above the AeroCom model mean global dust AOD; however, the broadness of the AeroCom model distribution implies that a global dust AOD greater than 0.035 would be required for statistically significant disagreement at the 95% confidence level (i.e. $p < 0.05$; in this case $p = 0.63$). Relative to the dust AOD from the four models used in this study (see Figure 2), all lie within $1\sigma$ of the observational estimate. The average global dust AOD estimates from each satellite instrument are remarkably similar (MODIS Aqua: $0.030 \pm 0.004$, MODIS Terra: $0.030 \pm 0.004$, MISR: $0.030 \pm 0.006$). This is partially owing to the AERONET bias correction that decreases the AOD from all satellite instruments and brings them into closer agreement. The AERONET bias correction suggests that the satellite AOD is generally biased high in dusty regions, based on the available data for comparison in the regions of interest (see Figures S1 to S3). On an annual basis, the observationally-constrained global dust AOD varies between 0.028 and 0.032, with good agreement in the interannual variability in dust AOD derived from the three instruments (Figure 4). The dust AOD is similar for years between 2004 and 2006 before increasing in 2007 and peaking in 2008, largely driven by a sharp increase across the Middle East (Yu et al., 2015). The AeroCom model simulations are representative of the year 2000; therefore, some of the difference between the global dust AOD in this study and that from the AeroCom study may derive from the interannual variability. However, the annual global dust AOD for each year equals or exceeds the AeroCom mean and median. Figure 3 and Figure 4 suggest that the global dust AOD from models in this study is in general agreement with the observational AOD constraints; whereas the models from the AeroCom study show more diversity. We note that the global dust AOD masks important regional differences that are discussed in Section 4.3. Furthermore, considerable uncertainty remains on the dust loading despite the similarities in global dust AOD as a result of compensating differences in dust emission, optics and aerosol size distribution assumptions (e.g. Albani et al., 2014; Balkanski et al., 2007; Cakmur et al., 2006). A follow-on study indicates that both the abundance and extinction efficiency of dust are underestimated in models and better constrains these factors to improve estimates of the dust impact on global climate (Kok et al.*, under review*).

## 4.2 Uncertainties in the observational estimate of global dust AOD

Table 1 summarizes the uncertainties considered in this study, both in terms of potential bias to the global dust AOD and the contribution to the standard deviation of the estimate (0.005). The latter is quantified by assessing the reduction in the spread of the global dust AOD PDF when the uncertainty for a factor is omitted. The leading uncertainty arises from the AERONET bias correction ($\alpha$, Eqn. 1). The bias correction yields a decrease in the global dust AOD of 10% and brings the estimates from each satellite instrument into close agreement, but the uncertainty on the bias correction accounts for over half of the ultimate uncertainty on the global dust AOD. The instrument retrieval errors contribute 5% of the uncertainty, whereas estimating the seasonal satellite AOD from a limited number of retrievals contributes 13% of the total uncertainty. The difference in regional-to-global dust AOD scaling from models and the difference in non-dust AOD from models yield $\pm6\%$ and $\pm8\%$ uncertainty, respectively, on the estimated global dust AOD. The latter uncertainty is primarily a consequence of higher non-dust AOD in MERRAero than the other three models and therefore a lower estimate of dust AOD. The uncertainty from non-dust AOD may not be symmetrical about the mean and is discussed further in Section 4.4 and in Supplementary

Materials. The regional-to-global scaling factor (β, Eqn. 3) is strongly dependent upon the dust lifetime within the model and ranges from 1.20 to 1.45, a lower scaling factor indicative of less dust far from source and therefore a shorter dust lifetime. The uncertainty from the regional-to-global scaling may not be symmetrical about the mean if the model dust lifetime estimates are biased low, as analysis of dust outflow into the mid-Atlantic suggests (see later discussion).

Other factors that are explored, but not encompassed by the uncertainty estimate on the global dust AOD, are the impact of spatial and temporal sampling biases in the satellite data (e.g. overpass timing and frequency, regions of persistent cloud, high latitudes), cloud filtering of satellite AOD retrievals, and inclusion of the Gulf of Guinea region. These are also included in Table 1 and discussed below.

      Satellite retrieval of AOD is only possible in clear sky conditions and at locations that fall within the satellite swath;
therefore, the observed dust AOD will not take into account the effect of dust present before or after the satellite overpass, and in the presence of clouds. We assess the impact of this sampling bias by processing the AOD from the 4 models in the same way as the satellite-retrieved AOD, including masking the daily AOD data where no satellite retrieval is available. By comparing the modeled dust AOD with and without masking, we determine that the impact of satellite sampling upon the global dust AOD estimate is minimal, < 1% for the MODIS instruments and +1.3% for MISR. Masking does however increase
the uncertainty in the dust AOD estimate by 7% when sampling is based on MODIS and 50% when sampling to the sparser MISR retrievals. Because the masking effectively removes cloudy regions, the very small change in the modelled global dust AOD indicates that there is no obvious bias in the global dust AOD when including regions within cloudy air masses, relative to clear-sky only. We also calculated GEOS-Chem global dust AOD after masking columns that have >50% cloud cover in any grid box, based on MERRA reanalysis. This causes the global dust AOD to increase by 2%, relative to when no masking
is used, indicating that the difference between clear-sky and all-sky dust AOD is small. However, we acknowledge that poor representation of clouds in the reanalysis meteorology or potential satellite misclassification of heavy dust loading as cloud (Darmenov and Sokolik, 2009) could lead to a stronger perceived relationship between dust loading in cloudy and clear sky conditions.

      The sun-synchronous orbit of the Terra and Aqua satellites results in overpass at similar morning and afternoon local
time, respectively, each day. Therefore, a significant daily cycle in the AOD would create a bias in the inferred daily AOD. For all dust-influenced AERONET sites, we compare the 10.00 – 12.00 LT and 14.00 – 16.00 LT AOD to the daily AOD (calculated from all available retrievals within the daytime) between 2002 and 2012. We find that, on the days with AE > 0.4 at the AERONET sites used in this study, the AOD during the morning and the afternoon are closely related to the daily AOD, deviating by < 2% on average. This is in agreement with Smirnov et al., (2002) that found AOD varied diurnally by less than
10% at dust-influenced AERONET sites. When the satellite retrieved AOD is bias corrected to the daily AOD, rather than the AOD at time of overpass (as done here), we find that the dust AOD is 4% lower (see Table 1).

      By filtering MODIS daily AOD 1° x 1° retrievals that contain more than 80% cloudy Level-2 pixels we find that the AOD drops considerably in the Mid-Atlantic, Gulf of Guinea and the Arabian Sea (Figure 5). This leads to significantly different estimates for the dust AOD in certain regions (Figure 6). The largest impact is seen in the Mid-Atlantic where the

dust AOD declines by 40% on average when filtering for clouds. The models also decrease when the equivalent masking is applied, but only by 20% on average in this region. This suggests that the filtering preferentially removes higher dust AOD cases, but the association of high dust AOD with cloudy regions is stronger in the observations than in the model. Similarly, reductions in dust AOD of up to 30% in winter and spring are produced by cloud filtering in the Gulf of Guinea. In the southern part of the Arabian Peninsula and the Arabian Sea the summertime peak in dust AOD is decreased by 30% by filtering pixels with more than 80% cloud cover. Cloud filtering of Level-2 retrievals is generally considered conservative in Collection 6 and misclassification is more common for thin cirrus than cumulus cloud decks (Levy et al., 2013; Remer et al., 2012). Therefore, removal of large regions in which high dust loading is associated with cumulus and stratus clouds may introduce an erroneous negative bias. It is also possible that high AOD retrieval in cloudy regions is the result of hygroscopicity and 3-D cloud effects (Koren et al., 2007; Marshak et al., 2008; Quaas et al., 2010). Indeed, it has been shown in studies using AERONET that AOD can increase dramatically between clouds and may be mistakenly screened as cloud (Eck et al., 2014). While this is a legitimate AOD enhancement, we cannot expect the global models with >100 km resolution using assimilated meteorology to reproduce enhancements from near-cloud hygroscopic growth or 3-D cloud effects on scattering. The observational-estimate of dust AOD provided in our analysis does not include the extra cloud filtering; we rely on the screening provided as part of the MODIS retrieval, rather than arbitrarily filtering the cloud-cleared product. However, the AERONET bias correction does decrease the AOD substantially, especially in the Mid-Atlantic (up to a 20% decrease), and so may partially account for the higher AOD associated with cloudy regions.

The regions defined around the South American, Southern African and Australian deserts and the outflow cover relatively large areas that are only intermittently affected by dust (see Figure 1). This may increase the likelihood of misattribution of non-dust AOD as dust AOD. We find that including those regions in the analysis does not have a significant impact on the global dust AOD, increasing it by 2%, although it causes a 6% increase in the uncertainty. In contrast, including the Gulf of Guinea region increases the dust AOD by +6% and increases the uncertainty by 9%. The dust AOD in the Gulf of Guinea region is consistently higher in the observational estimate than the models owing to a combination of persistent cloud cover, high biomass burning emissions in winter that are not always captured by the models, and a lack of dust towards the equator in the models that may result from too efficient convective wet removal. To prevent an artificially high bias in the global dust AOD, we do not explicitly evaluate the Gulf of Guinea region in our estimates beyond the assessment of uncertainty in Table 1. This region is still accounted for in the global dust AOD via the regional-to-global dust AOD scaling (the 14 remaining regions account for 77% - 87% of the global dust AOD, depending on the model).

**4.3 Comparison of modeled and observed regional dust AOD**

Model dust emissions are often tuned to a specific annual global emission mass (Fairlie et al., 2007; Huneeus et al., 2011) or scaled to a global AOD inferred from assimilations (Mahowald et al., 2006; Rasch et al., 2001). The annual global dust AOD derived from the models in this study show encouragingly similar interannual variability to the observationally-constrained estimates (see Figure 4). However, tuning the models globally will not necessarily produce the right spatial and

seasonal distribution. Here we use the observational constraints developed in this study to highlight regional and seasonal discrepancies between models and observations in an effort to isolate potential errors that affect multiple models. We compare the interannual variability globally and the seasonal dust AOD aggregated over broad regions for each of the models with the observational estimates from each satellite instrument (Figure 7a). We also compare the climatological seasonal dust AOD
from each model with the range of the observational dust AOD for each region (Figure 7b). We provide regional disaggregation of these results in Figures 8 and 9 and summarize the seasonal observational dust AOD for each region in Table 3.

Broadly, in Figure 7 we see that the models, except MERRAero, overestimate the amount of dust AOD over Africa with respect to the satellite estimates. The models generally over-emphasize winter or spring dust at the expense of summer. This is especially the case for GEOS-Chem (highlighted in Ridley et al., 2014; see Fig. S4 therein) and for CESM, and likely a
consequence of the lack of convectively-driven dust emissions that will be somewhat alleviated by new parameterizations (e.g. Pantillon et al., 2016). Switching the dust scheme in GEOS-Chem to a new parameterization that does not rely on an explicit source function (Kok et al., 2014a, 2014b) does not alleviate the seasonality issue in Africa, suggesting that the poor performance relative to the other models is likely the result of meteorology rather than the dust parameterization. Isolating the dust AOD in sub-regions, we find that the models overestimate dust in the North Africa, West Africa, and Bodele/Sudan
regions, while better matching the dust AOD in the mid-Atlantic outflow region, although there is significant variability between the four models (see Figure 8 and 9 for the sub-regions).

In the models, dust AOD over North Africa is greater than observed and dust AOD over the mid-Atlantic is often lower than observed (see Figure 8), even when extra cloud filtering of the satellite retrievals is included. This yields a ratio of the dust AOD over Africa to that over the Mid-Atlantic of $3.46 \pm 0.25$ for the models and $2.30 \pm 0.16$ based on observations ($2.62$
$\pm 0.16$ and $1.63 \pm 0.08$, respectively, with cloud filtering applied). The predominant direction of long-range transport of dust is across the Atlantic; therefore, the models are likely to be removing African dust too rapidly during transport. This is unlikely to be the result of too much dust mass concentrated at large particle sizes that sediment out rapidly, based on comparison between observed and modeled size distributions (Kok, 2011a). Instead, it may stem from the vertical distribution and mixing in the planetary boundary layer that can increase dry and wet removal through proximity to the surface and co-location with
precipitating clouds, respectively. Indeed, the choice of boundary layer mixing scheme can have a significant impact on long-range dust transport (Jin et al., 2015). The GEOS-Chem model dust lifetime over the Atlantic was shown to be 25-50% shorter than inferred from MODIS and primarily controlled by wet removal, that dominates over dry deposition in the mid-Atlantic region (Ridley et al., 2012). It is unclear whether this bias is connected to a poor representation of the Saharan Air Layer (SAL) at present model resolution or an unidentified source of systematic bias. Higher resolution simulations will be required to
capture the structure of the SAL, which can act as a conveyor for dust across the Atlantic. Excessive removal of dust will bias modeled dust lifetime low and result in a conservative observational dust AOD estimate because of the regional-to-global scaling employed in this study. The range of model dust lifetimes results in 13% to 23% of the global dust AOD coming from regions outside of those considered explicitly in this study. This constitutes a ±6% (0.0018) uncertainty in the observational global dust AOD estimate (Table 1); therefore, based on the comparison of dust AOD across the mid-Atlantic it is plausible

that the actual global dust AOD is towards the upper limit of this uncertainty bound. While the model representation of transport and deposition of mid-Atlantic dust may not be a major factor in the global dust AOD, it could have important implications for the simulation of hurricane genesis and nutrient deposition in the Amazon.

The models consistently underestimate AOD over Asian desert regions throughout most seasons (Figure 8). The low bias is present across all models and in all seasons except fall, when dust AOD is relatively low. The greatest divergence between models and observations occurs in spring AOD peak at the Taklamakan desert and in summer peak in the Thar desert, located between India and Pakistan. Only CESM and MERRAero capture the seasonality in the Thar region. Enhanced summertime coarse mode AOD retrieved at AERONET sites in Karachi and Jaipur, located on either side of the Thar desert, indicates that the models are likely missing dust emissions rather than the observational estimate being biased high. The low bias in modeled dust AOD is less pronounced in the Gobi desert, where GEOS-Chem and WRF-Chem appear to capture the observed spring peak in dust AOD. However, Figure 8 indicates that there is considerable uncertainty between the observational estimates in the Gobi Desert.

In the Middle East there is a slight low bias in the models relative to the observational dust AOD, through a combination of a substantial low bias in the Southern Middle East region, and a slightly high bias in the Northern Middle East and Kyzyl Kum regions. We find general agreement between the modeled and observed seasonality, with a spring peak in the Northern Middle East region and summer peaks in the Southern Middle East and Kyzyl Kum regions. However, all but MERRAero overemphasize summer dust at the expense of winter in the Kyzyl Kum region. CESM produces too much dust in summer, relative to other seasons, driven by high dust AOD between the Southern Middle East and Kyzyl Kum regions (the Gulf of Oman) that is present, but weaker, in the satellite observations.

Considering the southern hemispheric regions, our analysis indicates that the simulated dust AOD is comparable in Australia and lower than observed in South Africa and South America. However, the uncertainty in the observational dust AOD is too large to draw quantitative conclusions about the model representation of dust in those regions.

Throughout the comparison we find that MERRAero generally provides dust AOD that agrees better with the observational estimates, both in seasonality and magnitude, relative to the other models. This is expected as the MERRAero simulation involves assimilation of MODIS AOD retrievals (Buchard et al., 2015) and is therefore not independent from the observations to which we are comparing. Furthermore, MERRAero is also produced at a higher resolution than the other models (0.625° x 0.5°) which may further contribute to better representation of dust emissions owing to more spatially resolved surface winds. However, the magnitude of the global dust AOD provided by the MERRAero simulation is the lowest of the four models (0.027) and is 0.003 lower than the observationally-constrained estimate presented here (within the 1σ uncertainty bound). The total global AOD for all species in MERRAero is 5% - 15% higher than the other models, while dust accounts for a smaller fraction of the AOD in MERRAero (20%) than the other models (24% - 26%). This can be interpreted in two ways: either the contribution of dust to the total AOD is conservative in MERRAero, or the observationally-constrained estimates are biased high owing to a persistent low bias of non-dust AOD in three of the four models.

## 4.4 Discussion of the remaining uncertainties

We endeavoured to account for the uncertainties and biases involved in estimating the global dust AOD from observations; however, uncertainties remain that are difficult to quantify within this study. Potential sources of bias stem from (1) the model non-dust AOD, (2) the model regional-to-global AOD scaling, (3) treatment of particle morphology and mineralogy in models and in the satellite retrievals. These may present additional biases in the observational estimate and contribute to the discrepancies between models and observations.

We use multiple models to represent the uncertainty in non-dust AOD. However, the non-dust AOD in all models may be systematically biased high or low, which would bias the observational estimate of the dust AOD low or high, respectively. Comparison between modeled and observed AOD at the AERONET sites and MAN ship locations does suggest a low bias in the modeled total AOD in some of the regions considered, although there is no clear systematic bias in the models (see Figures S5 – S9). Comparison of model and AERONET AOD in low and high dust cases (using the model dust AOD to discriminate) suggests that two of the models are biased high and two biased low (Figure S4). Overall, the ensemble of models appears to underestimate the non-dust AOD; correcting this results in a 7% decrease in the global dust AOD estimate (0.028). However, the uncertainties involved in this method are such that we do not include the bias correction in our final estimate (see Supplementary Materials).

Modeled dust AOD is used as a scaling factor to determine the global dust AOD from the regional observational estimates. We use multiple models to represent the uncertainty, but there may be a systematic bias present, rather than the ±6% uncertainty presented (Table 2). If the over-zealous removal of dust in models, highlighted in the mid-Atlantic, is a global phenomenon then the models would predict too much dust in the source regions relative to downwind and yield a low regional-to-global scaling factor. Similarly, dust emissions schemes currently used in the models are unlikely to reproduce emissions where vegetation cover is variable and will not represent dust from agricultural regions (Ginoux et al., 2012). If those emissions are substantial, then it is possible that tuned emissions in models overestimate emissions from large, permanent dust sources to compensate for the lack of agricultural emissions, which could partially explain model bias towards African emissions.

Some of the discrepancy between the dust AOD from models and observations is likely born out of simplifications in representing particle morphology and minerology and the resulting impact on the AOD. The models in this study assume a globally fixed refractive index for dust and either spherical or spheroid particle shapes. We do not quantify the uncertainty from mineralogy and morphology here; however, several studies have shown the influence of refractive index and shape upon the derived optical and radiative properties (e.g. Balkanski et al., 2007; Kalashnikova and Sokolik, 2004; Scanza et al., 2015). Scanza et al. (2015) estimate a reduction of approximately 6% on the global dust AOD when accounting for spatially varying mineralogy in the Community Atmosphere Model (CAM-5). Particle morphology and minerology may also present a general bias in AOD retrievals as well as the models. Simplified particle shape modeling during retrieval has been shown to cause underestimation of AOD from space-based retrievals and overestimation from ground-based observations (Kalashnikova and Sokolik, 2002). Similarly, strongly absorbing dust can result in underestimation of the AOD, although improvements in

MODIS Collection 6 have been shown to alleviate this (Hsu et al., 2013). The impact on the observational estimate of dust AOD will be dependent upon the specific assumptions made by the MODIS and MISR retrievals, both of which take particle non-sphericity into account but using different methodologies (see Sections 2.1 and 2.2 and references therein). Finally, potential biases exist via erroneous filtering of thick dust plumes during the retrieval (Baddock et al., 2016).

**5 Conclusions**

To provide an observational constraint for the global dust AOD we use three satellite retrievals of AOD over a 5-year period, AERONET observations to correct biases in the satellite retrievals, and speciated aerosol AOD from four global chemical transport models to separate the contributions of dust and non-dust AOD. Throughout the analysis we use bootstrapping to retain a robust estimate of the uncertainty on the dust AOD. We determine the global dust AOD to be 0.030
$\pm$ 0.005 (1$\sigma$), with nearly three-quarters (73%) of the ensemble members in this study yielding a larger dust AOD than the mean of the 15 AeroCom models (0.028 $\pm$ 0.011) and all combinations greater than the AeroCom model median (0.023). The observational estimate narrows the likely range of dust AOD by half from that presented by the model estimates. The observational dust AOD is constructed as seasonal averages for 5 years (2004 - 2008) across 15 regions, providing a dataset with which the broad performance of model dust schemes can be evaluated (summarized in Table 3, with further data available
by request to the author).

All four models used in this study are within the one standard deviation uncertainty of the global mean observational estimate. However, it is essential to evaluate models on regional and seasonal scales, at which we find considerable differences. Using the regional and seasonal estimates of dust AOD, we highlight four general discrepancies between the models and observations: (1) the dust AOD across most of North Africa is overestimated in the models; (2) the Asian and Middle-Eastern
deserts are underrepresented overall, (3) modeled seasonality varies considerably between models but generally overestimates winter and spring dust at the expense of summer in Africa, and overestimate fall dust at the expense of spring in Asian deserts, and (4) removal of dust exported across the Atlantic appears to be too strong in the models, which may indicate a systematic underestimation of dust lifetimes. We have used the observationally-constrained estimate of dust AOD to isolate specific regions in which the models disagree with the observations. However, the underlying mechanisms for the discrepancies are
unclear and may be driven by the assumed physical characteristics of the surface, by the representation of surface wind, by the subsequent transport and deposition, or likely a combination of all factors. Further research in the areas highlighted in this work is expected to improve model simulations, and hence future estimates of the radiative, human health, and biosphere interactions of mineral dust.

## Acknowledgements

This work was supported by NASA under grant NN14AP38G. J.F.K. acknowledges support from the National Science Foundation (NSF) under grant 1552519. Chun Zhao is supported by the U.S. Department of Energy (DOE) as part of the Regional & Global Climate Modeling (RGCM) program.

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

**Tables**

| Region | AERONET bias correction and correlation coefficient | | | | | |
| --- | --- | --- | --- | --- | --- | --- |
| | MODIS (Aqua) | | MODIS (Terra) | | MISR | |
| | NMB | r | NMB | r | NMB | r |
| Mid-Atlantic | $0.82 \pm 0.07$ | 0.93 | $0.86 \pm 0.06$ | 0.93 | $0.84 \pm 0.04$ | 0.93 |
| N. Africa | $0.91 \pm 0.22$ | 0.79 | $1.01 \pm 0.12$ | 0.80 | $0.99 \pm 0.09$ | 0.83 |
| Gulf of Guinea | $0.90 \pm 0.10$ | 0.84 | $1.01 \pm 0.11$ | 0.83 | $0.92 \pm 0.10$ | 0.82 |
| W. Coast | $1.04 \pm 0.17$ | 0.80 | $1.04 \pm 0.19$ | 0.83 | $0.94 \pm 0.10$ | 0.87 |
| Mali/Niger | $0.98 \pm 0.17$ | 0.86 | $0.95 \pm 0.17$ | 0.87 | $0.93 \pm 0.12$ | 0.83 |
| Bodele/Sudan | $1.01 \pm 0.19$ | 0.68 | $0.99 \pm 0.19$ | 0.76 | $0.91 \pm 0.07$ | 0.70 |
| N. Mid-East | $0.81 \pm 0.10$ | 0.75 | $0.86 \pm 0.12$ | 0.73 | $0.95 \pm 0.08$ | 0.77 |
| S. Mid-East | $1.01 \pm 0.11$ | 0.77 | $1.06 \pm 0.11$ | 0.78 | $0.88 \pm 0.06$ | 0.84 |
| Kyzyl Kum | $1.02 \pm 0.29$ | 0.83 | $1.05 \pm 0.22$ | 0.82 | $1.19 \pm 0.10$ | 0.80 |
| Thar | $1.03 \pm 0.12$ | 0.84 | $1.04 \pm 0.15$ | 0.84 | $1.28 \pm 0.14$ | 0.79 |
| Taklamakan | $0.82 \pm 0.17$ | 0.66 | $0.98 \pm 0.21$ | 0.79 | $0.77 \pm 0.16$ | 0.47 |
| Gobi | $0.98 \pm 0.41$ | 0.54 | $0.90 \pm 0.42$ | 0.45 | $0.66 \pm 0.28$ | 0.73 |
| S. America | $0.85 \pm 0.15$ | 0.43 | $0.95 \pm 0.22$ | 0.18 | $0.56 \pm 0.15$ | 0.27 |
| S. Africa | $1.44 \pm 0.23$ | 0.73 | $1.71 \pm 0.27$ | 0.74 | $1.08 \pm 0.11$ | 0.89 |
| Australia | $1.02 \pm 0.32$ | 0.42 | $1.01 \pm 0.28$ | 0.43 | $0.92 \pm 0.10$ | 0.82 |

**Table 1 – Bias corrections applied to satellite AOD retrievals in each of the regions (see Figure 1) based on comparison with AERONET daily AOD between 2003 and 2013 (see Figures S1 – S3)**

| Source of uncertainty | Relative bias in global dust AOD | Relative contribution to uncertainty |
|---|---|---|
| Instrument retrieval uncertainty | <1% | +5% |
| Satellite retrieval of seasonal AOD | <1% | +13% |
| Model non-dust AOD | ±8% * | +6% |
| Model regional-to-global scaling | ±6% | +4% |
| AERONET bias correction (MODIS Aqua, MODIS Terra, MISR) | -10% (-4%, -10% -16%) | +56% (+64%, +73%, +50%) |
| *Satellite retrieval spatial sampling (MODIS Aqua, MODIS Terra, MISR)* | <1% (<1%, <1%, +1.3%) | -- (+7%, +7%, +50%) |
| *Satellite retrieval diurnal sampling (MODIS Aqua, MODIS Terra, MISR)* | -4% (-5%, -1%, -2%) | +2% (-1%, +2%, +3%) |
| *Cloud filtering (>80%)* | -13%** | <1% |
| Inclusion of S.H. desert regions | 2% | +6% |
| *Inclusion of Gulf of Guinea* | +6%** | +9% |

**Table 2 – Each source of uncertainty is assessed in terms of the impact upon the global dust AOD mean and standard deviation. The sign of the relative uncertainty indicates whether the uncertainty yields a bias about the average or is assumed symmetrical. For the model non-dust AOD and regional-to-global scaling the bias is defined as the difference between the upper and lower estimate of the global dust AOD when the source of uncertainty is isolated. Italicized uncertainties are explored but not incorporated into the global dust AOD uncertainty estimate. Values here are for correlated errors between neighboring 2 x 2.5 degree grid cells; assuming errors within a region are uncorrelated (i.e. a different number of sigma from the mean for each grid cell in an iteration) yields ~8x smaller uncertainty.**

**\* May not be symmetrical about the mean (see Supplementary Materials, Figure S4)**

**\*\* Relative to global dust AOD without AERONET bias correction**

| Region | DJF | | MAM | | JJA | | SON | |
|---|---|---|---|---|---|---|---|---|
| **Asia** | 0.114 | ±0.017 | 0.237 | ±0.017 | 0.191 | ±0.027 | 0.094 | ±0.021 |
| **Mid-East** | 0.119 | ±0.012 | 0.201 | ±0.017 | 0.250 | ±0.021 | 0.129 | ±0.014 |
| **Africa** | 0.167 | ±0.007 | 0.291 | ±0.012 | 0.298 | ±0.017 | 0.196 | ±0.015 |
| N. Africa | 0.118 | ±0.011 | 0.219 | ±0.010 | 0.207 | ±0.016 | 0.151 | ±0.016 |
| Mid-Atlantic | 0.064 | ±0.013 | 0.106 | ±0.008 | 0.143 | ±0.005 | 0.084 | ±0.006 |
| Mali/Niger | 0.257 | ±0.019 | 0.441 | ±0.022 | 0.462 | ±0.044 | 0.277 | ±0.023 |
| Bodele/Sudan | 0.191 | ±0.006 | 0.339 | ±0.023 | 0.310 | ±0.018 | 0.212 | ±0.021 |

| | | | | | | | |
|---|---|---|---|---|---|---|---|
| W. Coast | 0.180 | ±0.010 | 0.250 | ±0.019 | 0.365 | ±0.016 | 0.233 | ±0.022 |
| S. Mid-East | 0.123 | ±0.018 | 0.204 | ±0.021 | 0.330 | ±0.044 | 0.150 | ±0.020 |
| Kyzyl Kum | 0.115 | ±0.017 | 0.176 | ±0.026 | 0.154 | ±0.034 | 0.101 | ±0.018 |
| N. Mid-East | 0.112 | ±0.011 | 0.223 | ±0.011 | 0.164 | ±0.015 | 0.113 | ±0.019 |
| Thar | 0.130 | ±0.029 | 0.238 | ±0.033 | 0.319 | ±0.029 | 0.135 | ±0.037 |
| Gobi | 0.093 | ±0.022 | 0.192 | ±0.022 | 0.102 | ±0.035 | 0.047 | ±0.021 |
| Taklamakan | 0.119 | ±0.013 | 0.275 | ±0.027 | 0.171 | ±0.026 | 0.104 | ±0.011 |
| S. Africa | 0.097 | ±0.023 | 0.073 | ±0.022 | 0.059 | ±0.021 | 0.114 | ±0.040 |
| Australia | 0.022 | ±0.016 | 0.008 | ±0.009 | -0.005 | ±0.008 | 0.001 | ±0.023 |
| S. America | 0.020 | ±0.017 | 0.000 | ±0.013 | -0.012 | ±0.013 | 0.017 | ±0.013 |

**Table 3 – Observational estimates of the seasonal dust AOD in each region of Figure 1, averaged over 2004 - 2008. The first three rows show the seasonal dust AOD for the broad regions (grouped by color in Figure 1; Africa does not include the Gulf of Guinea region)**

**Figures**

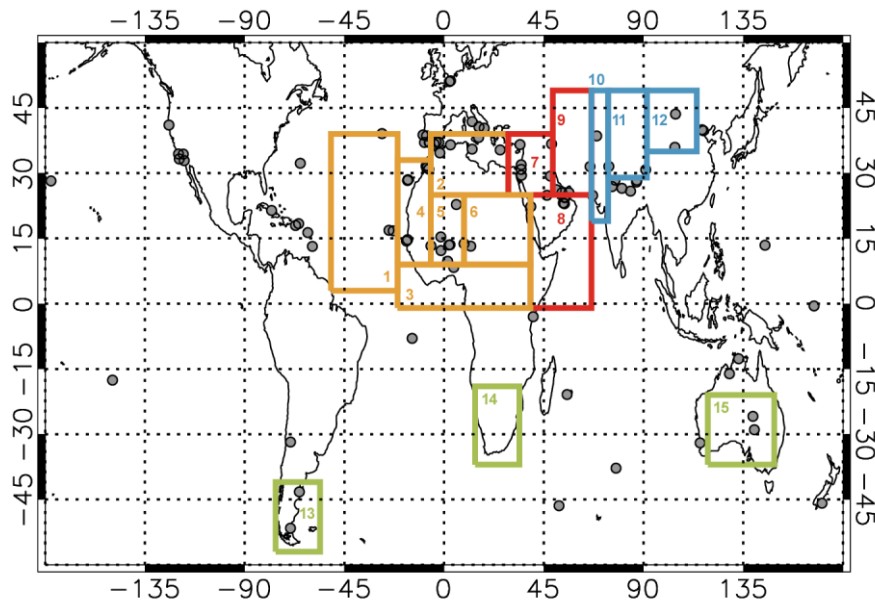

**Figure 1 – The 15 regions considered explicitly in this study are defined. Regions are grouped into African (orange), Middle Eastern (red), Asian (blue), and Southern Hemispheric (green). AERONET sites used to bias correct satellite AOD are indicated with (gray circles). The regions are identified as (1) Mid-Atlantic, (2) N. Africa, (3) Gulf of Guinea, (4) W. Coast, (5) Mali/Niger, (6) Bodele Depression and Sudan region, (7) N. Mid-East, (8) S. Mid-East, (9) Kyzyl Kum, (10) Thar, (11) Taklamakan, (12) Gobi, (13) S.**
10 **America, (14) S. Africa, and (15) Australia.**

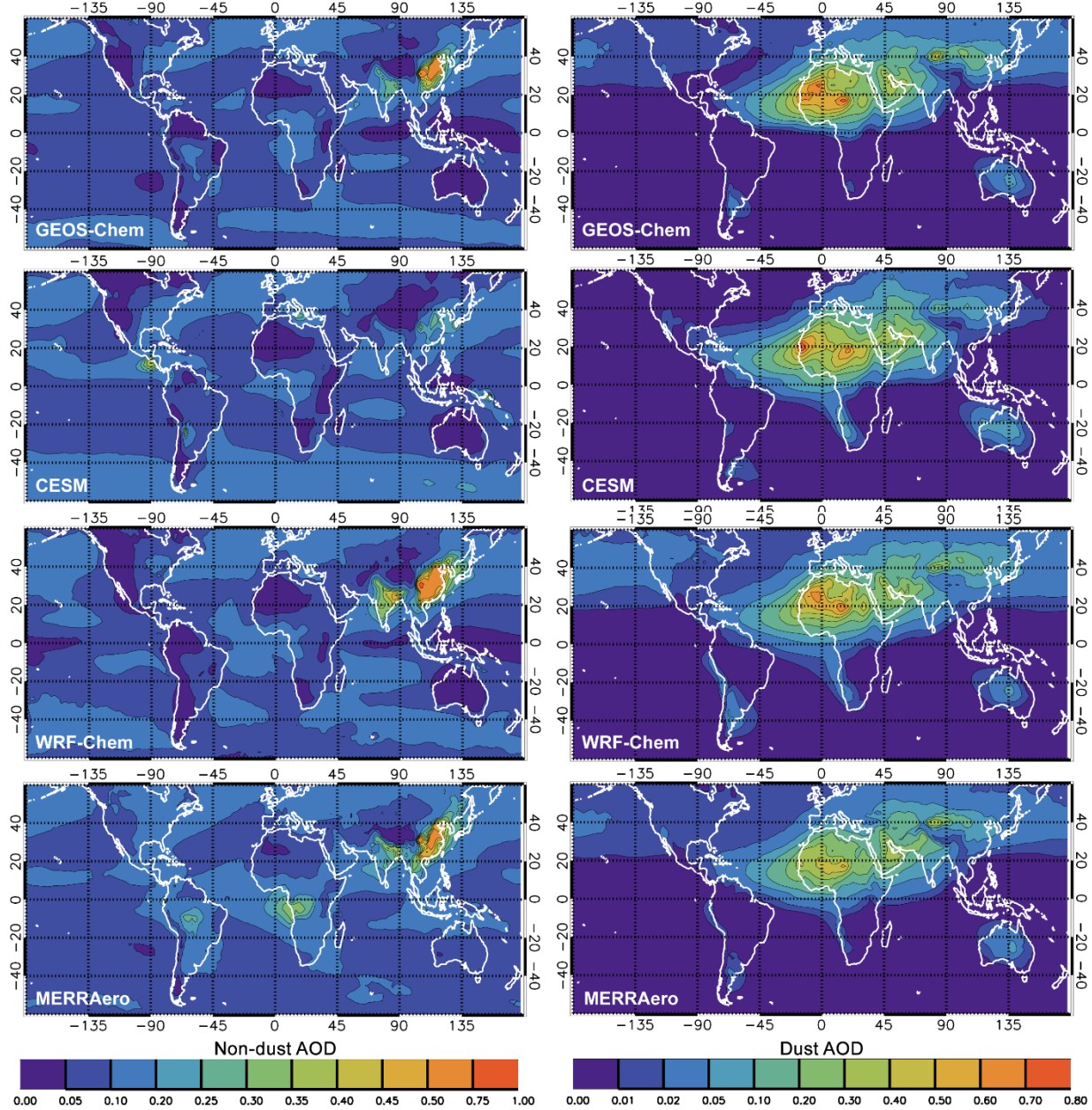

**Figure 2 – Annual non-dust AOD (left) and dust AOD (right) at 550 nm for the four models used in this study. Data is averaged over 2004 – 2008.**

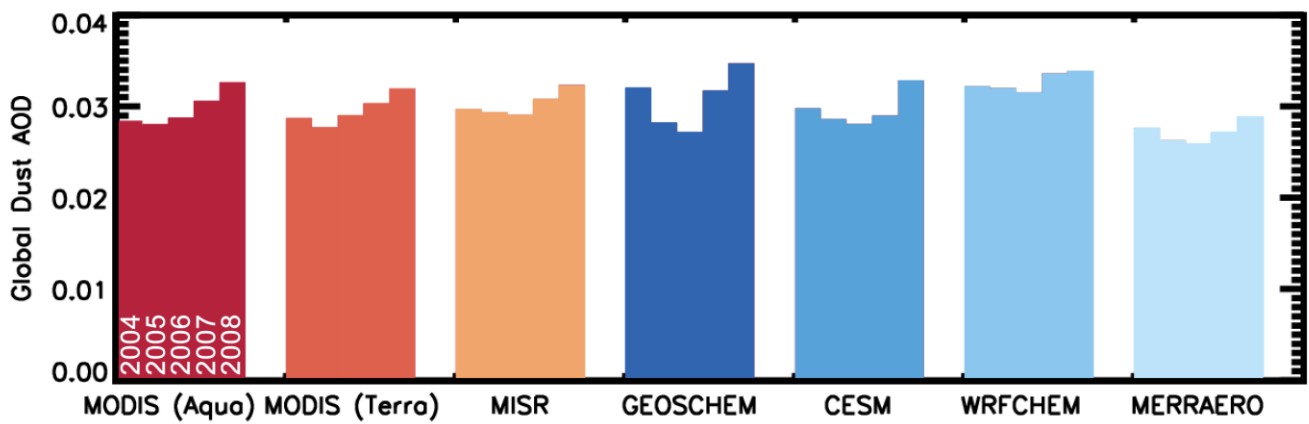

**Figure 3 – The global dust AOD adapted from Huneeus et al. (2011) for 14 AeroCom models (vertical black lines) and the associated PDF (solid black line), mean (dashed black line) and the AeroCom median model (dotted black line) are shown along with the global dust AOD from the four models used in this study (vertical blue lines). The PDF of the observationally-constrained dust AOD estimate of this study (red) with the associated mean (dashed red line) is shown on the bottom axis. The PDF of the observationally-constrained dust AOD derived from each of the satellite instruments is shown (red hues) with the individual ensemble members (vertical red hue lines).**

**Figure 4 – Annual global dust AOD for 2004 – 2008 derived from the three satellite instruments (red hues) and from the four models (blue hues). The annual dust AOD from the satellite instruments is an average of the ensemble members for that instrument.**

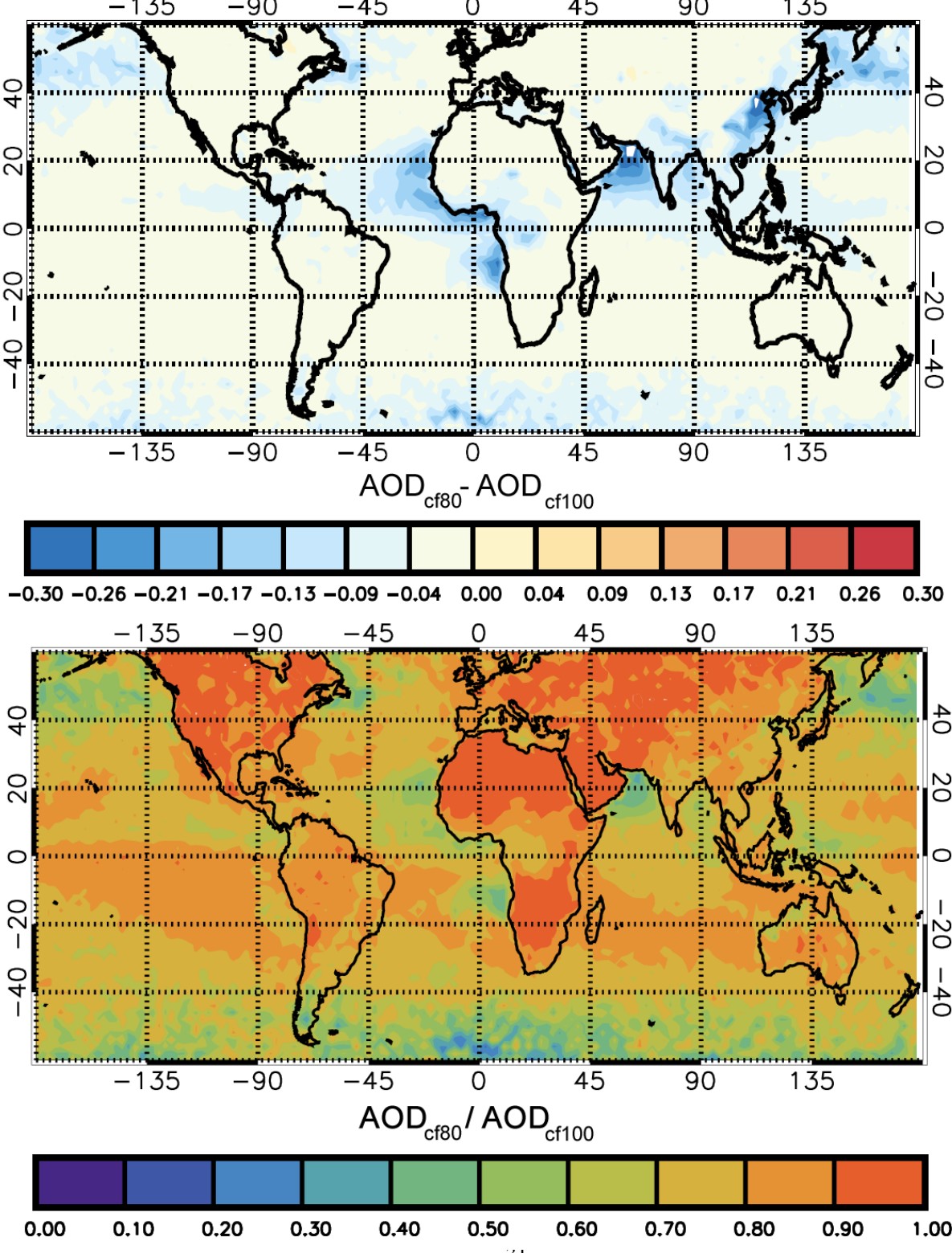

**Figure 5 – The absolute change (top) and fractional change (bottom) in the annual MODIS Aqua AOD (averaged over 2004 - 2008) when applying a filter to remove any Level-3 data that contains more than 80% cloudy Level-2 pixels.**

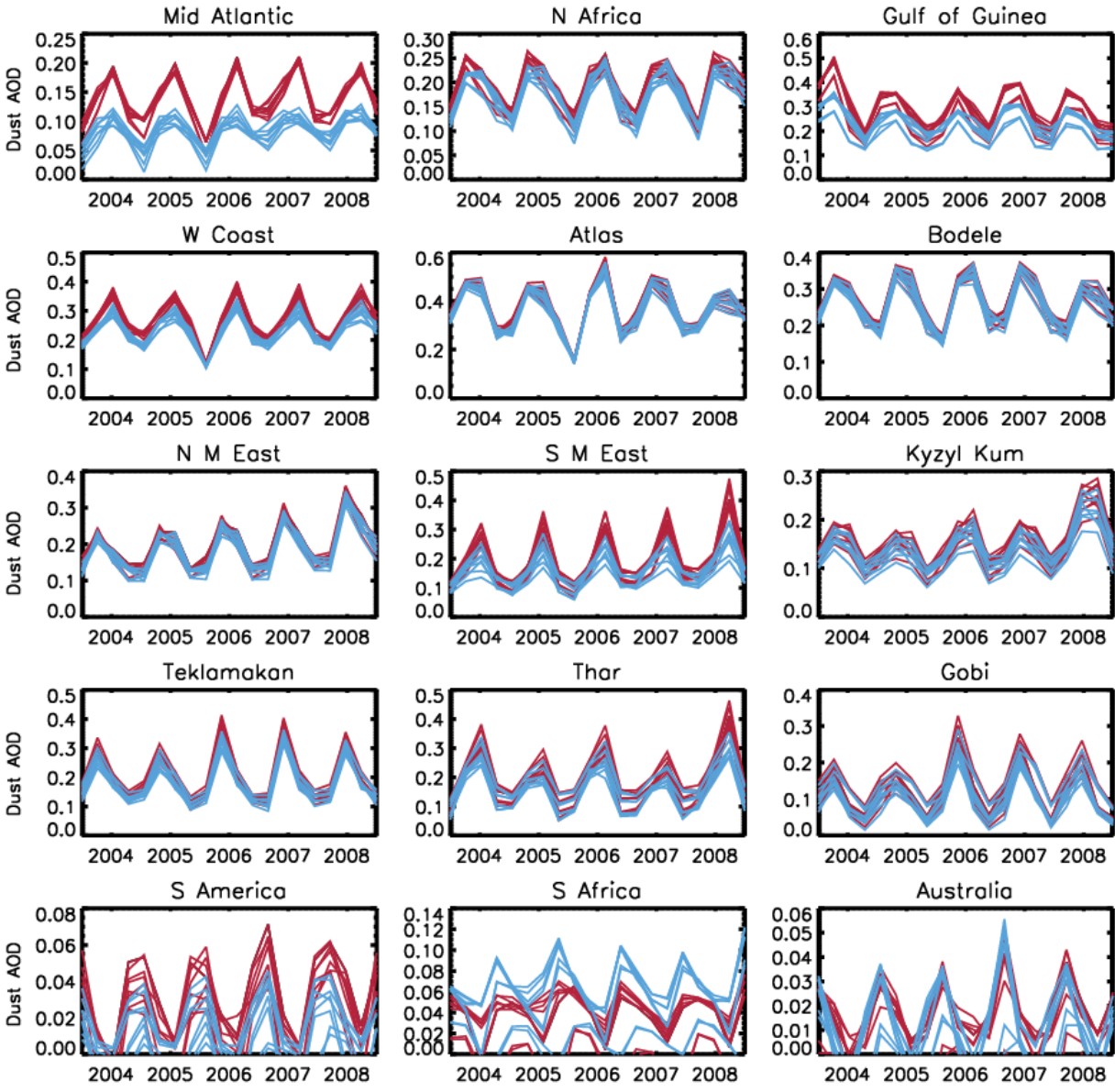

**Figure 6 – Observational dust AOD from MODIS Aqua and Terra with (red) and without (blue) filtering of 1° x 1° daily regions with over 80% cloud cover. Each line corresponds to a different combination of satellite and model when calculating the dust AOD, indicating the uncertainty. Results are shown without any bias correction from AERONET.**

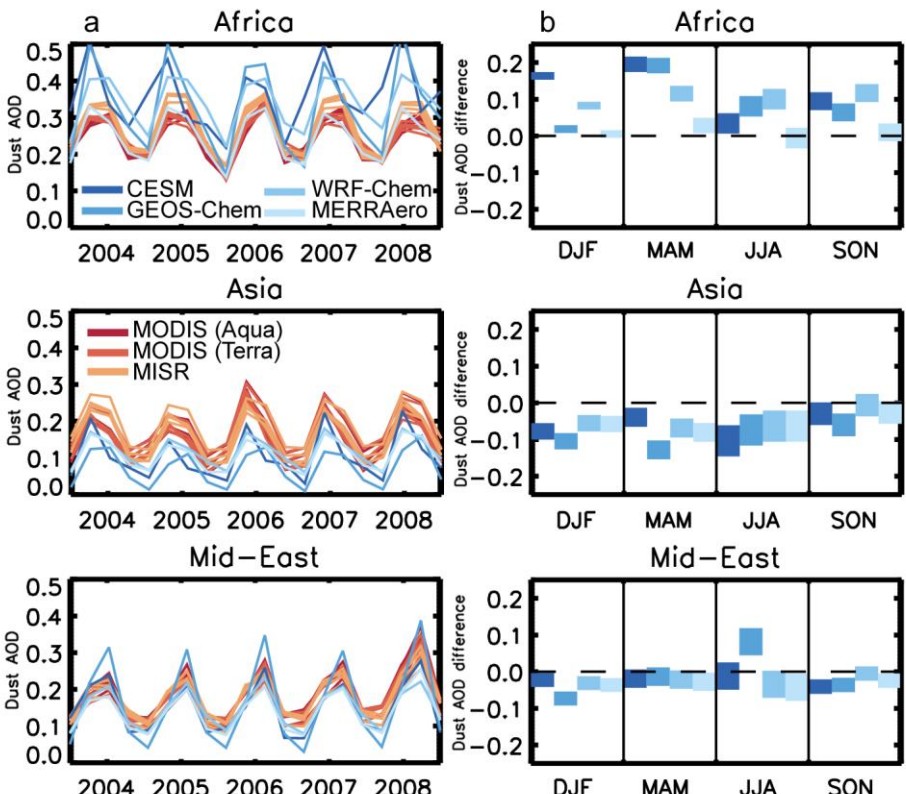

**Figure 7 – (a)** Seasonal dust AOD between 2004 and 2008 for observational-based estimates (red hues; each of the 12 lines represents a different ensemble member) and for the models (blue hues). **(b)** To isolate the seasonality, the difference between model and observational-based seasonal dust AOD, averaged over 2004-2008, is shown. Bar thickness indicates the range of the observational-based estimates for each season, deviation from zero (dashed line) indicates the bias in model seasonal dust AOD relative to the observations. The regions are based on area-weighted averages over the subset of regions defined in Figure 1, except Africa, which does not include the Mid-Atlantic region (shown separately in Figure 8 and 9 with other sub-regions).

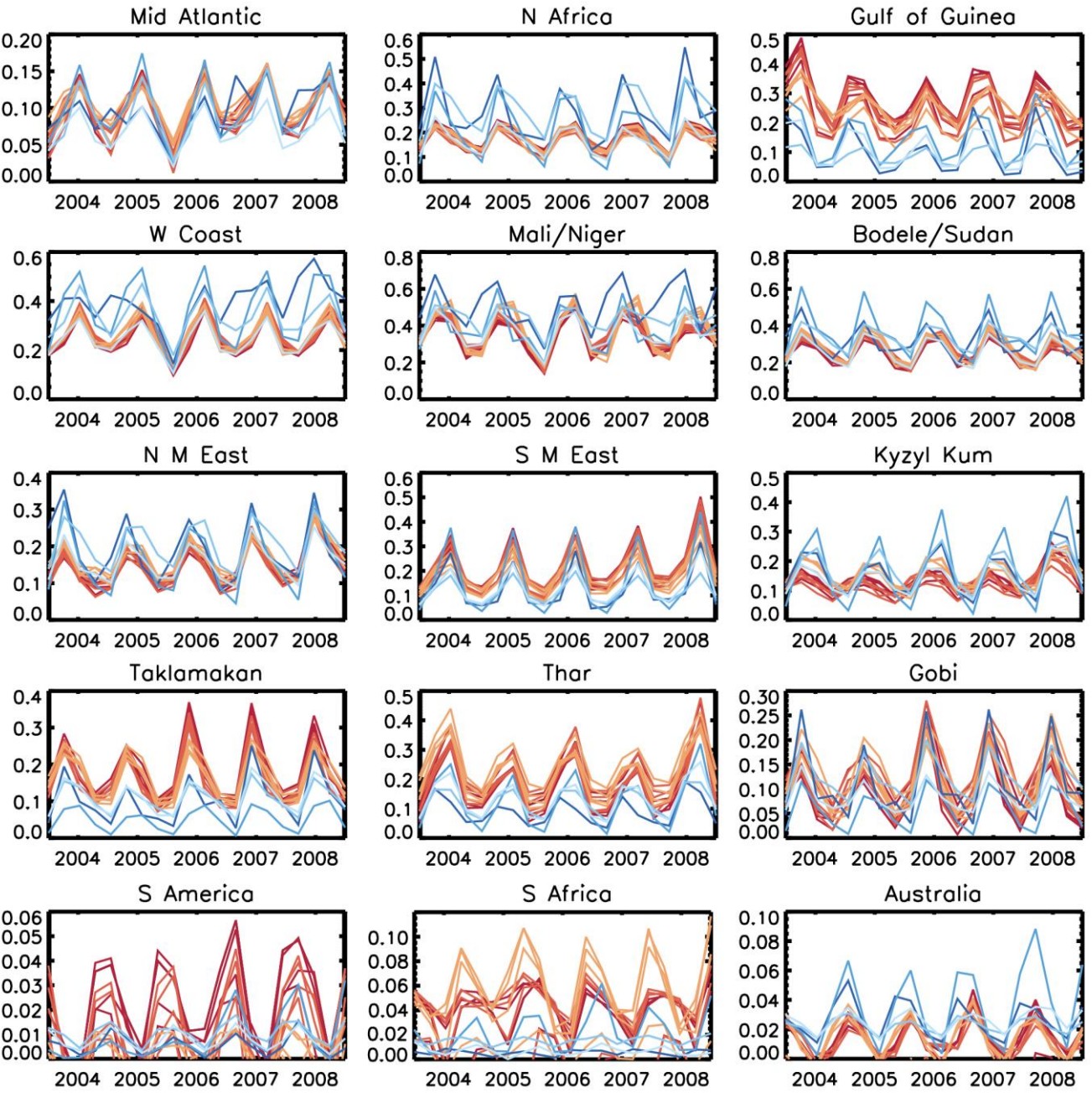

**Figure 8 – Same as Figure 7a but for each individual region (in Figure 1). Observational dust AOD is shown for multiple realizations of MODIS Aqua, MODIS Terra and MISR (dark to light red). Models are GEOS-Chem, CESM, WRF-Chem and MERRAero (dark to light blue).**

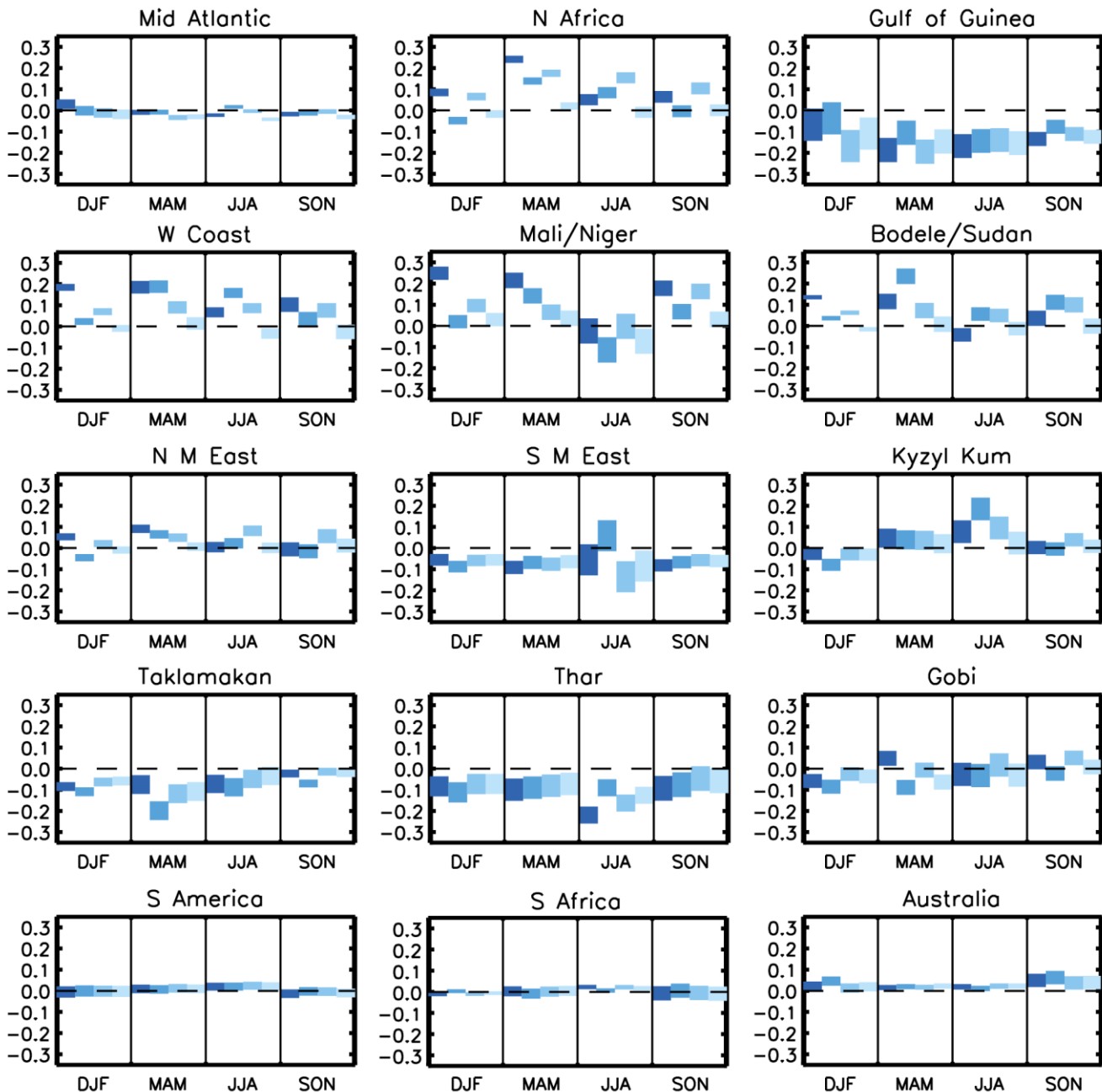

**Figure 9 – same as Figure 7b but for individual regions (in Figure 1). Models are GEOS-Chem, CESM, WRF-Chem and MERRAero (dark to light blue).**

