# Peer review of "An observationally-constrained estimate of global dust aerosol optical depth"

_Atmospheric Chemistry and Physics, 2016_

## Short Comment (SC1) · Comments on Ridley et al. · 10 Jun 2016

This is a potentially really important paper, with a sound methodology, for the most part. The issues come with the error analysis, which appears to substantially underestimate the errors. The paper also fails to provide context with previous studies. If these issues are fixed, the paper is likely to be extremely influential.

The main errors associated with knowing the dust aod come from: 1. errors with the retrieval algorithm, 2) spatial and temporal hetereogeneity in dust distribution, 3) spatial and temporal variability in dust composition and/or shape, 4) errors in detecting dust

versus other aerosols or clouds.

The authors seem to deal fairly well with the 4th of these, but seem to underestimate the errors in the other three. Please discuss the issues with the retrievals and all the problems with the retrieval algorithms. Are the algorithms making the same assumptions about dust properties? That would then add another error, which will be difficult to assess by just comparing different datasets. For example, if they assume all dust is one optical property, or spherical, or at particular altitudes, etc. Please describe these sources of errors.

In the comparison of the MODIS, MISR and aeronet, what is the rms error? This error represents a combination of the spatial and temporal variability as well as errors in the retrieval algorithms, and needs to propagate into the error in your final estimate. As it stands, only the mean bias propagates into your error estimate, which will underestimate your errors. If I look at Moon et al., 2015, the error bar on individual retrievals in MISR are at least 30%: how can you claim smaller error than that in your results? You seem to be assuming that these errors will average out, but this seems unlikely and this assumption would have to be justified.

You include a comparison of AOD across all sites in the world, with all types of aerosols. How does this comparison over just dusty regions compare? Is it better or worse, please explain.

Dust is not homogeneous in chemical composition, size and thus optical properties, but the retrieval algorithms assume that they are. You should explicitly discuss this point, and you could bound the error from mineralogy using Scanza et al., 2015, which suggest for the CAM5, the impact of spatially varying optical properties depending on mineralogy is 0.002 out of 0.033 aerosol optical depth or about a 6% error (1 sigma).

Then it would seem you would need to add all these errors to the total estimated error, without letting them cancel each other, and then it seems likely that you will get a reasonable value.

The last comment is to consider how this estimate differs from previous model/data comparisons (e.g. Cakmur et al., 2007; Albani et al., 2014 or Balkanski et al. 2007). There are two main differences. Here the primary spatial and temporal variability relationships come from the satellite remote sensing data vs. model results in those papers. And secondly, because the first two papers include comparisons to concentration and deposition data. To understand how important the second is, please provide a comparison of your 'constrained' AOD-implied concentration and deposition to available datasets. This can be done very simply, but just using, for example, the GEOS-CHEM dust AOD to deposition to surface concentration relationships, and your inferred AOD at that grid box. That will allow you to do a very simple comparison and show that indeed, your approach is (probably) fairly consistent with the other datasets. It probably won't be completely consistent, since none of the models seem to be able to match the AOD, concentration and deposition data the same time.This information could be added to the supplemental material and referenced briefly in the text.

---

## Referee Comment (RC1) · Anonymous Referee #4 · 16 Jun 2016

The study combines estimates of AOD from satellite and sun-photometer (AERONET) observations. The authors evaluate the statistical uncertainty of dust AOD calculated from model simulations against in-situ observations. The manuscript is well written and scientifically sound.

General comment:

Why do you scale the model AOD from regional to global (page 8 & 9, Eqn 3)? The general scaling approach does not consider the regional variability in soil properties (determining dust emission fluxes), meteorological drivers, size distributions (affecting AOD and life time), etc. What is the motivation for ignoring these factors despite knowing that they affect on dust concentrations and dust properties? Are the results after

scaling still representative? Please consider including some words on how meaningful the scaling approach is.

Related to that, can global averages of dust AODs considered as an appropriate measure for model skills with regard to dust distribution? Regional errors may equal out and thus a global average may be misleading. As also pointed out in the result section, dust varies strongly with regions and depends on the model skills for the regions.

Furthermore, on the one side you are arguing with global averages of AOD (i.e. abstract and conclusion), on the other side you are suggesting that regional means are the more appropriate measure. It sounds somewhat inconsistent. Please clarify.

Specific comments:

p4 l11 remove parenthesis for reference Kok et al.

p4 l24 remove parenthesis for reference Albani et al.

p5 l7 remove parenthesis for reference Gong, 2003

p5 l14 remove parenthesis for reference Fast et al., 2006

p5 l14 remove parenthesis for reference Barnard et al., 2010

p7 l4 "man" should be "main"

p7 l19 should be MERRAero to be consistent

p9 l14 It appears a bit odd to me to have one of the co-authors cited as "personal communications". Maybe omit the "personal communication" part and only provide the "manuscript in preparation" part?

p9 l23 "Eqn. 3" to be consistent

p9 l27 "Eqn. 1" to be consistent

p11 l30 As the naming of the regions are erroneous on the figures (see below), please

check if it's correct in the text.

p13 l1 Please consider shifting "(the Gulf of Oman)" to line 26 where the Kyzyl Kum region was mentioned first.

Fig. 1 something went wrong with assigning geographical names to the numbering of the areas. Area number 5 is definitely not the Atlas Mountain region. Maybe confound with the Adrar des Iforas Mountain region? Similarly, the Bodele Depression covers the Sudan, too. Please clarify.

Fig. 7, 9, 10 Base on the numbering issue appearing in Fig. 5, there may be a consequent mis-naming of the Atlas region. Please check.

Fig. 7, 9, 10 Taklamakan
* * *

---

## Referee Comment (RC2) · Anonymous Referee #3 · 16 Jun 2016

In this manuscript the authors present a global reconstruction of dust AOD based on satellite data and sun photometer retrievals, using sun photometer data to correct satellite bias and various model simulations to separate the regional contribution of dust from other aerosols. This is a really nice manuscript with some good ideas and a dataset the has the potential to be a widely cited reference. Because of this potential it is necessary to be extra careful, though. The authors have followed previous methodology, including the weaknesses. I'd like to see these addressed before I support the publication of this manuscript.

Major Comments: 1. One of my major concerns is the use of the different emission schemes in different models. This will have an impact on the calculation of the dust

[Figure]

AOD (eq. 1). How much of the model-ensemble uncertainty is due to different emission schemes and how much due to inter-model variability?

2. I haven't found an explanation why the AOD reconstruction is limited to the 15 regions. Why do you not reconstruct AOD over the whole globe and show it on a map (e.g. using a yearly median)? You can still only calculate the correction factor using the dust-dominated regions.

3. The assumption on which the correction factor are based seem weak to me. It is true that most of the global dust AOD is dominated by the North African and East Asian region. But this doesn't mean that you only need to concentrate on a few region, but that the spatial distribution is no Gaussian. In fact, if you look at a histogram of a snapshot in time you will probably find that dust is spatially log-normally distributed. My suggestion to the authors is to look at the spatial distribution of the satellite and sun photometer data and if it's lognormal, try to take the logarithm of all initial AOD data such that it is spatially normally distributed and rethink their calculations (especially equations 1-3 and Figure 2) and discussion from that perspective.

4. Global means make sense for GHG but not for aerosols. Talking about a global mean AOD is meaningless. It gives you absolutely no information about what the AOD could be on any point on Earth. I know everybody's doing it and there's a weak argument that can be made for inter-paper comparison's sake. But this manuscript has the potential to be a widely cited reference and it has the means to provide data for more regionally-based comparisons in the future. Figure 4 looks very fancy but gives very little useful information. Maybe in addition to Figure 4 that compares with previous papers you could prepare a synthesis figure or table with which people writing papers in the future can easily compare their results (something like figure 9 but less messy – no offense to figure 9).

Minor comments:

The references to air quality and health seem out of place in this manuscript. There is

no need to mention these aspect if they are not discussed anywhere.

Page 2, lines 7-9: I don't know if that's a mistake in the original Huneeus paper, but if you give the median because the distribution is not Gaussian, then you shouldn't give the standard deviation, which is a parameter in the Gaussian distribution. Chapters 2.1, 2.2, 2.3: I would appreciate it if the description of errors was consistent between the three instruments. Page 6&7, lines 32-7: Looking at the data in Figure 2 I would guess that the data is not normally distributed. The choice of a linear regression to calculate the bias between AERONET and satellites is therefore doubtful. See my major comment 3. Page 8, Eq.1: In my experience, aerosol concentrations, loads, and therefore AOD are not normally distributed in space. The mean AODs calculated here may not be representative of the central tendency in each region. See major comment 3. Page 10 line 3: AE<0.4 Figure 2: In the MISR panel, there are values only for one of the two regressions. Also I can see only one regression line
* * *

---

## Referee Comment (RC3) · Anonymous Referee #5 · 23 Jun 2016

The manuscript describes a new potential tool for validation of mineral dust in global and regional models, based on a combination of remote sensing data and global climate models. The work is certainly of interest and could provide an additional useful tool to the modeling community. In general the methodology appears sound and the paper is well organized and written. A few minor revision are nevertheless needed in my opinion before the paper could be published.

Major comment

The construction of the global AOD dataset is the central part of this work. It stems mainly from remote sensing observations, form both satellites and ground-based AERONET stations. I think that too little information is provided regarding data pro-

cessing (e.g. temporal aggregation) and uncertainties in these types of observations and their relation to dust AOD.

Specific comments

2, 8-9: please add a reference here.

2, 14-16: Why PM2.5 in particular? You do not discriminate the size in your product.

2, 24: It would be useful to mention already here what is the general strategy of the work, and why you will use all of the following data from observations or model. Maybe add a table or a brief description in the text, so that the reader can already have a better idea of the role of each type of data in this paper.

3, 15: the usage of the angstrom exponent is not clear, please rephrase.

4, 24: this sentence is not clear; also the reference is missing from the list.

8, 20-24: How is your central estimate derived? Is it the mean of the distribution derived from the set of all possible combinations of models and satellite data depicted in Figure 4? Also, please describe more in detail how all the combinations were constructed in the previous section.

---

## Referee Comment (RC4) · A. Evan (Referee) · 25 Jun 2016

This manuscript describes a method of combining satellite and model data in order to estimate the global dust AOD (DAOD). The principal idea here is that models do a good job of simulating non-dust AOD, and satellites do a good job of retrieving the total AOD, so the difference between the two should be a good estimate of DAOD. While I applaud the authors on their creative effort, and the obviously massive amount of time undertaken to complete this work, I find there to be a couple of major issues with the methods that I suspect are contributing to a bias in their global DAOD estimate, and increase the uncertainty. Thus, I am suggesting a major revision.

Signed, Amato Evan

Major Comments 1. A major assumption of this method is that model DAOD is biased, but that model AOD is not. However, this assumption, at least the part about model AOD not having any systematic bias, isn't justified. The authors suggest that they are accounting for errors related to underestimation of the non-dust AOD by reporting their global DAOD with a 2-sigma uncertainty range (P13, L15). However, if the models systematically underestimate the non-dust AOD, this will induce a high bias in their reported global DAOD, and thus simply increasing the uncertainty range isn't really appropriate. We need to know if there is a bias, particularly because a low bias in modeled non-dust AOD would serve to push the hybrid global DAOD estimate closer to the aerocom mean, and possible closer to the MERRAaero estimate.

One could determine if such a bias exists by comparing histograms of AOD for the models and AERONET, over land regions and over-water regions where there is no dust (but there is smoke, anthro. aerosols, and marine aerosols). The difference in those histograms can be used to calculate a bias (which could be corrected) and uncertainty in the models' non-dust AOD. These errors can then be carried through to the final global DAOD calculation.

2. I am also very concerned about use of the models' spatial structure of DAOD (the horizontal pattern of long-term mean DAOD). In Eqn 2 the authors rely on the spatial structure of modeled DAOD in order to estimate their hybrid global DAOD. The implicit assumption is that while the models' may exhibit biases in the absolute value of DAOD, they do a good job of reproducing the long-term mean spatial structure. However, later on in the paper (P11, Section 4.3) the authors examine the signs of the difference between modeled DAOD and that from their hybrid method in Fig 9 (Africa, N Atl, Gulf of Guinea), suggesting that the models emit too much dust at the source to compensate for the fact that wet and dry deposition is far too strong. So on the one hand you are saying that the spatial structure of model DAOD is good (Eqn 2) and on the other hand it's not (Fig 9).

If your hypothesis is correct, that the models emit too much dust because deposition is

too strong, then Eqn 2 will introduce a bias into your global DAOD estimate depending on the relative fraction of regions (Fig 1) that are over dust emitting areas and those that are downwind. I think this means that because your regions in Figure 1 are overwhelmingly near or over dust sources, your final global DAOD estimate could be biased low? I'm not entirely sure... But the bottom line is that, given this bias in the spatial structure of dust from the models, there is an additional source of uncertainty in the global DAOD estimate, and potentially a bias, related to the distribution of the regions you choose (Fig 1). I'm not exactly sure how you can address this. Maybe add more over-water regions and redo the estimate only using over-water regions, the only using over-land regions, then using both (via Eqn 2)? Or maybe the way to address this potential bias/uncertainty is to recalculate global DAOD using an equal distribution of regions over dust sources and regions downwind of dust sources (also in Eqn 2).

3. Lastly, I think models report AOD even in the presence of 100% cloud cover. So, in the model, there could be an aerosol layer overlaying stratus clouds, and the model would save an AOD value. However, in the satellite world, there would be no AOD retrieval. Does this discrepancy induce a bias? Can you examine the model data (I guess you'd need daily or hourly output) to see?

Minor Comments

1. P7, L4: Spelling, "main" not "man"

2. Should alpha have a region superscript in Eqn 1?

3. P11, L25: Why would a lack of convectively driven dust emissions cause an overestimation of DAOD? Seems like it would be the opposite.

4. P7, L26: You write, "In the regions analyzed here the AOD is predominantly driven by dust aerosol, limiting the influence of the model non-dust AOD" but this simply isn't true. Region 1 (N. Atl) also has a big biomass burning contribution in the boreal winter. Regions 8 also has a contribution from anthro. aerosols from N. India during the dry
monsoon season. Same for region 10 (from Pakistan and Iran).

5. P10: Cloud filtering: Interesting that you are getting such a strong correlation between the two. Misclassification of optically thick dust as cloud may be pretty common, FYI.

---

## Short Comment (SC2) · 29 Jun 2016

This is a very nice work, which will provide a better constrained mean dust load and optical depth. Still, I wonder about some biases related to satellite data in general, and MODIS Deep Blue in particular. The authors note a lack of bias in MODIS AOD based on the scatter plot of daily values at AERONET sites. However, these sites are characterize by different aerosol environments and surface albedo. Uncertainties related to satellite retrieved AOD between sites will be different. In Figure 3 of Ginoux et al. (Rev. Geophys., 2012), you will notice very different biases between regions. For example Australia is biased high, while Africa is slightly biased low. Although this study was done with Collection 5.1, similar results are obtained with Collection 6, but with

much more reduced bias in Australia. My point is that there is very little information we can extract from your Figure 2. A better approach would be to also plot seasonal variation at dusty sites (e.g. Tamanrasset, Birdsville, Solar Village. Dunhuang, etc.). I am also concerned about your method of temporal average of observations. If you consider only days with retrievals you will have a high bias, as you discard all days with dust being washout and rainout (low dust). This will be also true for AERONET data. But, it is unclear which method you are using. None of the models simulate dust from agricultural regions or with dynamic vegetation. Their contribution is highly uncertain but may affect your results regionally. Finally, you are most likely using MODIS quality flag 3 (QA=3) aerosol products, as advised by Sayer et al. (2013). However, it is not a good choice over dust sources as clearly shown in Figure 1 of Baddock et al. (Geophys. res. Lett., 2015). This choice of QA=3 may induce a low bias, if you use all days rather than just days with QA=3. On the other hand, if you divide the sum of all valid AOD by the number of days with QA=3, you will again create a high bias. In fact, it may be very high in some areas. Take a look at the factor 10 difference of frequencies between QA=1 and QA=3 in Figure 1 of Baddock et al. (2015).

Hopefully this will help improve your results.

Paul Ginoux.
* * *

---

## Author Comment (AC1) · 27 Oct 2016

Response to Reviewers

We would like to thank all reviewers for their helpful comments and criticism on this work. We believe we have addressed the comments and made changes to the methodology and manuscript where possible. We now include supplementary figures and several of the figures in the manuscript have been updated.

Key changes include: • Analysis and statistics generated for log(AOD) rather than AOD • Instrument uncertainty included in the estimate • Regional bias correction of satellite data by AERONET • Uncertainty in bias correction propagated through

analysis • Marine Aerosol Network (MAN) data included • Supplementary figures of AERONET and satellite AOD histograms • Comparison of model AOD with daily AOD from MAN • Supplementary comparison with deposition flux

The key changes are that the global dust AOD is decreased from 0.033 to 0.030 and the uncertainty increased from 0.006 to 0.011 ($2\sigma$) as a result of considering instrument uncertainty and the uncertainty on the updated AERONET bias correction of the satellite retrievals. The observational estimate is hence closer to the AEROCOM model estimate. We believe that this better corrects for regional biases in the satellite retrievals while representing the inherent uncertainty in using limited in-situ measurements to apply correction factors over large regions. The regional estimates of seasonal dust AOD from the different satellite instruments are generally in closer agreement. The observational estimate is also brought closer to the MERRAero dust AOD; the previous discrepancy was of some concern because MERRAero assimilates MODIS AOD and may be expected to represent the dust AOD better than models without assimilation. The agreement between model and observational estimate improves over the mid-Atlantic, reducing (but not eliminating) the potential for systematically high dust removal in the models. While many of the quoted numbers change as a result of our reanalysis, all other conclusions remain essentially the same.

Please find the reviewer-specific comments and responses (blue italics) listed below.

Kind regards, David Ridley

Comments from Natalie Mahowald

This is a potentially really important paper, with a sound methodology, for the most part. The issues come with the error analysis, which appears to substantially underestimate the errors. The paper also fails to provide context with previous studies. If these issues are fixed, the paper is likely to be extremely influential.

Thank you. We hope that the additions we have made in response to your comments

and those of the referees have improved the paper significantly.

————

The main errors associated with knowing the dust aod come from: 1. errors with the retrieval algorithm, 2) spatial and temporal hetereogeneity in dust distribution, 3) spatial and temporal variability in dust composition and/or shape, 4) errors in detecting dust versus other aerosols or clouds. The authors seem to deal fairly well with the 4th of these, but seem to underestimate the errors in the other three. Please discuss the issues with the retrievals and all the problems with the retrieval algorithms. Are the algorithms making the same assumptions about dust properties? That would then add another error, which will be difficult to assess by just comparing different datasets. For example, if they assume all dust is one optical property, or spherical, or at particular altitudes, etc. Please describe these sources of errors.

We agree that the uncertainty has been underestimated. We now incorporate the retrieval uncertainty in the first stage of the bootstrapping process. When bootstrapping to create a seasonal mean AOD with standard deviation, we include the instrument uncertainty on the daily AOD, and combine the errors for each day used in the seasonal mean AOD. This does not alter the mean, but increases the standard deviation of the AOD at a specific location. However, we find that the final global dust AOD uncertainty increases by <5% because the uncertainty is dwarfed by other factors, primarily the uncertainty on the new regional AERONET bias correction and the uncertainty on the satellite seasonal AOD derived from bootstrapping (Table 2). We recognize that region-specific retrieval biases may exist and will be unaccounted for. This is now acknowledged in Section 4.4, a new section for unaccounted uncertainties.

The retrieval algorithm assumptions will affect how well spatial and temporal hetereogeneity in the dust distribution, composition and shape are accounted for. We now discuss the retrievals in more detail in Sections 2.1 and 2.2 for MODIS and MISR to show differences in assumptions:

"The MODIS retrieval algorithm uses a look-up table of surface reflectance for a set of simulated aerosol properties to determine the AOD that best represents the observed reflectance. For the Deep Blue retrieval, the most relevant to this study over dust-influenced regions, the assumed optical properties of the dust aerosol have a single-scattering albedo (SSA) between 0.87 and 1.0 for the look-up tables at 412 nm and 490 nm and a refractive index of 1.55 – 0.0i (at 670 nm). The Mie calculation uses an effective phase function, derived from comparison of the Sea-Viewing Wide Field-of-View Sensor (SeaWIFS) instrument retrievals with AERONET, over the ocean to account for non-sphericity. Different locations and loading conditions trigger changes in the wavelengths used in the retrieval, more information can be found in Hsu et al. (2004, 2013)"

"The MISR retrieval algorithm uses simulated TOA radiances using properties for eight particle types to determine the AOD. The optical properties of the two aerosol particle types corresponding to dust are calculated using the discrete dipole approximation and the T-matrix technique to account for particle non-sphericity (Kalashnikova et al., 2005; Martonchik et al., 2009)."

The broad bias correction of satellite-retrieved AOD to AERONET may account for some of the biases in the retrieval, although this correction is very uncertain and we are neglecting that the AERONET retrieval is not perfect. We now account for the large uncertainty in the AERONET bias correction, propagating that through to the global dust AOD estimate.

————————

In the comparison of the MODIS, MISR and aeronet, what is the rms error? This error represents a combination of the spatial and temporal variability as well as errors in the retrieval algorithms, and needs to propagate into the error in your final estimate. As it stands, only the mean bias propagates into your error estimate, which will underesti-mate your errors. If I look at Moon et al., 2015, the error bar on individual retrievals in

MISR are at least 30%: how can you claim smaller error than that in your results? You seem to be assuming that these errors will average out, but this seems unlikely and this assumption would have to be justified.

We wouldn't necessarily expect the retrieval uncertainty to be similar to the 30% of Moon et al. as we are averaging over longer timescales and larger regions so should be able to beat down the uncertainty. However, as before there may be unaccounted biases. As mentioned above, we have revised the bias correction of the satellite data to be regional, rather than global and have added the histograms with statistics of the AERONET-satellite retrieval comparisons in the supplementary materials (Figures S1-S3). We now assess an uncertainty to the bias correction by calculating the standard deviation in bias corrections for each year with sufficient (>100) coincident satellite and AERONET retrievals. This propagates the uncertainty in the bias correction through the analysis and now accounts for half of the uncertainty in the global dust AOD (Table 2).

——————

You include a comparison of AOD across all sites in the world, with all types of aerosols. How does this comparison over just dusty regions compare? Is it better or worse, please explain.

Our regional comparison of satellite and AERONET AOD should address this concern. We have included histograms of the daily AOD comparisons within each region in Figure S1 – S3.

——————

Dust is not homogeneous in chemical composition, size and thus optical properties, but the retrieval algorithms assume that they are. You should explicitly discuss this point, and you could bound the error from mineralogy using Scanza et al., 2015, which suggest for the CAM5, the impact of spatially varying optical properties depending on

mineralogy is 0.002 out of 0.033 aerosol optical depth or about a 6% error (1 sigma). Then it would seem you would need to add all these errors to the total estimated error, without letting them cancel each other, and then it seems likely that you will get a reasonable value.

Thank you. We now include reference to Scanza et al. and discuss the added uncertainties from mineralogy and morphology (among other factors) in Section 4.4. However, it is not clear if the difference in global dust AOD when including mineralogy in CAM-5 (a decrease from 0.033 to 0.031) from Scanza et al. (2015) would bias our observational estimate higher or lower. Combining this error with our current estimate yields only a small increase; therefore, we simply highlight this uncertainty in the extra Section 4.4, added to discuss the uncertainties and biases that may not be captured by this work.
* * *
The last comment is to consider how this estimate differs from previous model/data comparisons (e.g. Cakmur et al., 2007; Albani et al., 2014 or Balkanski et al. 2007). There are two main differences. Here the primary spatial and temporal variability relationships come from the satellite remote sensing data vs. model results in those papers. And secondly, because the first two papers include comparisons to concentration and deposition data. To understand how important the second is, please provide a comparison of your 'constrained' AOD-implied concentration and deposition to available datasets. This can be done very simply, but just using, for example, the GEOSCHEM dust AOD to deposition to surface concentration relationships, and your inferred AOD at that grid box. That will allow you to do a very simple comparison and show that indeed, your approach is (probably) fairly consistent with the other datasets. It probably won't be completely consistent, since none of the models seem to be able to match the AOD, concentration and deposition data the same time. This information could be added to the supplemental material and referenced briefly in the text.

Thank you for this suggestion. We have produced an estimate of the dust deposition following the method you suggest above and have included comparison with deposition network data in Figures S10 and S11. We note that the result is strongly dependent upon the model used (GEOS-Chem). Also, we are producing regional dust AOD estimates that are not conducive to comparison with clustered deposition measurements. Indeed, the correlation is only slightly better than for the GEOS-Chem model, and not significantly so – in part owing to the reliance on the model AOD distribution and the model relationship between dust AOD and dust deposition. For this reason, it is not easy to compare with previous model studies. We have made sure to reference this work so that the reader can dig deeper into the underlying assumptions in the models and how they affect the model representation of concentration and deposition as well as the AOD.

---

## Author Comment (AC2) · 27 Oct 2016

Response to Reviewers

We would like to thank all reviewers for their helpful comments and criticism on this work. We believe we have addressed the comments and made changes to the methodology and manuscript where possible. We now include supplementary figures and several of the figures in the manuscript have been updated.

Key changes include: • Analysis and statistics generated for log(AOD) rather than AOD • Instrument uncertainty included in the estimate • Regional bias correction of satellite data by AERONET • Uncertainty in bias correction propagated through

analysis • Marine Aerosol Network (MAN) data included • Supplementary figures of AERONET and satellite AOD histograms • Comparison of model AOD with daily AOD from MAN • Supplementary comparison with deposition flux

The key changes are that the global dust AOD is decreased from 0.033 to 0.030 and the uncertainty increased from 0.006 to 0.011 ($2\sigma$) as a result of considering instrument uncertainty and the uncertainty on the updated AERONET bias correction of the satellite retrievals. The observational estimate is hence closer to the AEROCOM model estimate. We believe that this better corrects for regional biases in the satellite retrievals while representing the inherent uncertainty in using limited in-situ measurements to apply correction factors over large regions. The regional estimates of seasonal dust AOD from the different satellite instruments are generally in closer agreement. The observational estimate is also brought closer to the MERRAero dust AOD; the previous discrepancy was of some concern because MERRAero assimilates MODIS AOD and may be expected to represent the dust AOD better than models without assimilation. The agreement between model and observational estimate improves over the mid-Atlantic, reducing (but not eliminating) the potential for systematically high dust removal in the models. While many of the quoted numbers change as a result of our reanalysis, all other conclusions remain essentially the same.

Please find the reviewer-specific comments and responses (blue italics) listed below.

Kind regards, David Ridley

Anonymous Referee #4 The study combines estimates of AOD from satellite and sun-photometer (AERONET) observations. The authors evaluate the statistical uncertainty of dust AOD calculated from model simulations against in-situ observations. The manuscript is well written and scientifically sound. Thank you for your comments, we hope to have covered your concerns below.

_________________-
* * *
Interactive
comment

General comment: Why do you scale the model AOD from regional to global (page 8 & 9, Eqn 3)? The general scaling approach does not consider the regional variability in soil properties (determining dust emission fluxes), meteorological drivers, size distributions (affecting AOD and life time), etc. What is the motivation for ignoring these factors despite knowing that they affect on dust concentrations and dust properties? Are the results after scaling still representative? Please consider including some words on how meaningful the scaling approach is.

The scaling to global dust AOD does rely on the global distribution of dust aerosol in the four models used, and will represent the regional variability in soil properties, meteorological drivers and size distributions to the extent that those models reproduce those properties. We are unable to account for potential biases that exist in all the models; however, the purpose of using four models is to both reduce the impact of these biases and to propagate their effect on the uncertainty in the DAOD by providing a range of scaling factors.

We derive the dust AOD over regions in which dust aerosol makes up a significant fraction of the total AOD to minimize errors from both retrieval uncertainty and model representation of non-dust AOD. Ideally we would derive the dust AOD in all regions to eliminate the need to use the models; however, the uncertainties prevent meaningful results in remote regions. We found that comparison of satellite retrievals of AOD with the Marine Aerosol Network (MAN) showed poor correlation and bias in remote locations. We have added the following text to clarify this in the manuscript (pg 8):

"We focus on regions in which the dust AOD often dominates to reduce potential errors from biases in modeled non-dust AOD."

This is followed by examples of key uncertainties in the non-dust AOD (e.g. regions influenced by biomass burning).

————————————-

Related to that, can global averages of dust AODs considered as an appropriate measure for model skills with regard to dust distribution? Regional errors may equal out and thus a global average may be misleading. As also pointed out in the result section, dust varies strongly with regions and depends on the model skills for the regions. Furthermore, on the one side you are arguing with global averages of AOD (i.e. abstract and conclusion), on the other side you are suggesting that regional means are the more appropriate measure. It sounds somewhat inconsistent. Please clarify.

We agree that there is limited use for global dust AOD. However, this is a common metric used to assess models and is presented here to allow comparison with model estimates. Because of the limited use we have provided specific information on the seasonality and the magnitude of the dust AOD in different regions. We believe that including both the global dust AOD and the more detailed regional interpretation of the results is a reasonable framework. We address this in opening paragraph of the Section 3.4 on regional dust AOD and have added the following statement to the conclusions:

"...it is essential to evaluate models on regional and seasonal scales, at which we find considerable differences."

Specific comments:

————————-

p4 l11 remove parenthesis for reference Kok et al. Done

————————-

p4 l24 remove parenthesis for reference Albani et al. Done

————————-

p5 l7 remove parenthesis for reference Gong, 2003 Done

————————-

p5 l14 remove parenthesis for reference Fast et al., 2006 Done

———————————-

p5 l14 remove parenthesis for reference Barnard et al., 2010 Done

———————————-

p7 l4 "man" should be "main" p7 l19 should be MERRAero to be consistent Done

———————————-

p9 l14 It appears a bit odd to me to have one of the co-authors cited as "personal communications". Changed

———————————-

Maybe omit the "personal communication" part and only provide the "manuscript in preparation" part? Done

———————————-

p9 l23 "Eqn. 3" to be consistent Done

———————————-

p9 l27 "Eqn. 1" to be consistent Done

———————————-

p11 l30 As the naming of the regions are erroneous on the figures (see below), please check if it's correct in the text. Thank you, these have been corrected throughout the text.

———————————-

p13 l1 Please consider shifting "(the Gulf of Oman)" to line 26 where the Kyzyl Kum region was mentioned first. Here we are discussing the Gulf of Oman as the region between the Southern Middle East and Kyzyl Kum desert regions. Therefore, we believe the reference to the Gulf of Oman should stay in its current location.

————————-

Fig. 1 something went wrong with assigning geographical names to the numbering of the areas. Area number 5 is definitely not the Atlas Mountain region. Maybe confound with the Adrar des Iforas Mountain region? Similarly, the Bodele Depression covers the Sudan, too. Please clarify. Thank you, the Atlas mountains region was mis-labelled and has been corrected to Mali/Niger

————————-

Fig. 7, 9, 10 Base on the numbering issue appearing in Fig. 5, there may be a consequent mis-naming of the Atlas region. Please check. Fig. 7, 9, 10 Taklamakan

————————-

These have been corrected to better represent the regions: Mali/Niger and Bodele/Sudan, and the Taklamakan spelling used throughout.

---

## Author Comment (AC3) · 27 Oct 2016

Response to Reviewers

We would like to thank all reviewers for their helpful comments and criticism on this work. We believe we have addressed the comments and made changes to the methodology and manuscript where possible. We now include supplementary figures and several of the figures in the manuscript have been updated.

Key changes include:  c Analysis and statistics generated for log(AOD) rather than AOD

 c Instrument uncertainty included in the estimate

 c Regional bias correction of satellite data by AERONET

 c Uncertainty in bias correction propagated through analysis

 c Marine Aerosol Network (MAN) data included

 c Supplementary figures of AERONET and satellite AOD histograms

 c Comparison of model AOD with daily AOD from MAN

 c Supplementary comparison with deposition flux

The key changes are that the global dust AOD is decreased from 0.033 to 0.030 and the uncertainty increased from 0.006 to 0.011 ($2\sigma$) as a result of considering instrument uncertainty and the uncertainty on the updated AERONET bias correction of the satellite retrievals. The observational estimate is hence closer to the AEROCOM model estimate. We believe that this better corrects for regional biases in the satellite retrievals while representing the inherent uncertainty in using limited in-situ measurements to apply correction factors over large regions. The regional estimates of seasonal dust AOD from the different satellite instruments are generally in closer agreement. The observational estimate is also brought closer to the MERRAero dust AOD; the previous discrepancy was of some concern because MERRAero assimilates MODIS AOD and may be expected to represent the dust AOD better than models without assimilation. The agreement between model and observational estimate improves over the mid-Atlantic, reducing (but not eliminating) the potential for systematically high dust removal in the models. While many of the quoted numbers change as a result of our reanalysis, all other conclusions remain essentially the same.

Please find the reviewer-specific comments and responses (blue italics) listed below.

Kind regards, David Ridley

Anonymous Referee #3 In this manuscript the authors present a global reconstruction of dust AOD based on satellite data and sun photometer retrievals, using sun photometer data to correct satellite bias and various model simulations to separate the regional contribution of dust from other aerosols. This is a really nice manuscript with some good ideas and a dataset the has the potential to be a widely cited reference. Because of this potential it is necessary to be extra careful, though. The authors have followed previous methodology, including the weaknesses. I'd like to see these addressed before I support the publication of this manuscript.

We thank the reviewer for their comments and generally positive view of the research. We appreciate the points raised and believe that the major concerns have been addressed, detailed below.

––––––––––––-

Major Comments: 1. One of my major concerns is the use of the different emission schemes in different models. This will have an impact on the calculation of the dust AOD (eq. 1). How much of the model-ensemble uncertainty is due to different emission schemes and how much due to inter-model variability? Without the ability to plug the same emission scheme into each of the models this is difficult to assess. Through the ensemble of satellite instrument and model combinations we can separate the impact of model diversity in non-dust AOD from the model regional-to-global scaling. This is presented in Table 2, where we assume that the bias is symmetrical around the estimate.

––––––––––––-

2. I haven't found an explanation why the AOD reconstruction is limited to the 15 regions. Why do you not reconstruct AOD over the whole globe and show it on a map (e.g. using a yearly median)? You can still only calculate the correction factor using the dust-dominated regions.

Our rationale for using the 15 regions is to isolate the regions that are more influenced by dust than other aerosol species to minimize the effect of mis-categorizing errors in

non-dust AOD from the model as dust AOD. For example, the large AOD from pollution over East China is not always fully represented by the models and would result in a significant bias in the dust AOD inferred over this region. We therefore exclude regions such as this, Europe, and remote areas far from dust sources. Furthermore, through comparison between satellite retrievals and AERONET and the Marine Aerosol Network (MAN) we find that the retrievals are much less reliable in some locations, such as the Southern Ocean, and therefore unlikely to produce useful results. Lastly, the bootstrapping process requires many iterations of the dust AOD calculation at each location and is time-consuming. We have added this rationale to the manuscript as follows (pg 8):

"We focus on regions in which the dust AOD often dominates to reduce potential errors from biases in modeled non-dust AOD."

This is followed by examples of key uncertainties in the non-dust AOD (e.g. regions influenced by biomass burning).

——————-

3. The assumption on which the correction factor are based seem weak to me. It is true that most of the global dust AOD is dominated by the North African and East Asian region. But this doesn't mean that you only need to concentrate on a few region, but that the spatial distribution is no Gaussian. In fact, if you look at a histogram of a snapshot in time you will probably find that dust is spatially log-normally distributed. My suggestion to the authors is to look at the spatial distribution of the satellite and sun photometer data and if it's lognormal, try to take the logarithm of all initial AOD data such that it is spatially normally distributed and rethink their calculations (especially equations 1-3 and Figure 2) and discussion from that perspective.

We agree that the AOD is usually log-normal, thank you for this observation. We have repeated the analysis assuming the AOD is log-normally distributed when aggregating daily AOD retrievals into a seasonal mean with a standard deviation. Interestingly, we

find that this has a relatively small impact on the results, reducing the global dust AOD only slightly. However, this is certainly a better way to represent the statistics and have updated the manuscript accordingly.

We did use AERONET data from all stations when calculating the correction factor, not just those within the 15 regions (see pg6 line 29). However, based on comments from reviewers we have revisited the bias correction methodology and altered this to provide bias corrections for each region, based on the AERONET sites in that region (and using the Marine Aerosol Network where relevant). Histograms of the distribution of AOD for the satellite retrievals and for AERONET within each region are shown in Figure S1 to S3.

_______________-

4. Global means make sense for GHG but not for aerosols. Talking about a global mean AOD is meaningless. It gives you absolutely no information about what the AOD could be on any point on Earth. I know everybody's doing it and there's a weak argument that can be made for inter-paper comparison's sake. But this manuscript has the potential to be a widely cited reference and it has the means to provide data for more regionally-based comparisons in the future. Figure 4 looks very fancy but gives very little useful information. Maybe in addition to Figure 4 that compares with previous papers you could prepare a synthesis figure or table with which people writing papers in the future can easily compare their results (something like figure 9 but less messy – no offense to figure 9).

We understand the concern with using the global average AOD and its limitations. However, it is necessary to compare with the modeled estimates of Huneeus et al. (2011) and we find that the global dust AOD metric is useful for discussion of the factors leading to uncertainty in the observational estimate. We highlight the importance of regional assessment in Section 4.3:

"...tuning the models globally will not necessarily produce the right spatial and seasonal distribution. Here we use the observational constraints developed in this study to highlight regional and seasonal discrepancies between models and observations in an effort to isolate potential errors that affect multiple models."

Regarding comparisons with future work: we intend for the regional data to be available for future studies to compare against; however, we now provide a summary (Table 3) that gives our observational estimate of average seasonal dust AOD in the regions considered to facilitate quick comparison.

––––––––––––

Minor comments: The references to air quality and health seem out of place in this manuscript. There is no need to mention these aspect if they are not discussed anywhere.

We respectfully disagree. We believe it is important to highlight the influence of dust on air quality when AOD retrievals are used in assessing surface PM2.5 in poorly monitored regions, such as Africa, as part of studies informing World Health Organization assessments (e.g. van Donkelaar et al., 2006; Evans et al., 2013).

––––––––––––

Page 2, lines 7-9: I don't know if that's a mistake in the original Huneeus paper, but if you give the median because the distribution is not Gaussian, then you shouldn't give the standard deviation, which is a parameter in the Gaussian distribution. The "AEROCOM median" was a combination of the models in the AEROCOM analysis and treated as a separate model. This has been clarified by referring to it as the AEROCOM "model median".

––––––––––––

Chapters 2.1, 2.2, 2.3: I would appreciate it if the description of errors was consistent between the three instruments. We have homogenized the error format

[Figure]
* * *
Page 6&7, lines 32-7: Looking at the data in Figure 2 I would guess that the data is not normally distributed. The choice of a linear regression to calculate the bias between AERONET and satellites is therefore doubtful. See my major comment 3. Hopefully the response to comment 3 addresses this point.
* * *
Page 8, Eq.1: In my experience, aerosol concentrations, loads, and therefore AOD are not normally distributed in space. The mean AODs calculated here may not be representative of the central tendency in each region. See major comment 3. Hopefully this was addressed in response to the major comment. The manuscript has been updated to reflect the use of log-normal distributions.
* * *
Page 10 line 3: AE<0.4 Figure 2: In the MISR panel, there are values only for one of the two regressions. Also I can see only one regression line This figure has been removed, following suggestions to improve the bias correction of satellite data to AERONET observations, and no longer applies a split regression for MISR. Information on the bias correction of the satellite retrievals is now presented in Table 1 and Figures S1 – S3.

---

## Author Comment (AC4) · 27 Oct 2016

Response to Reviewers

We would like to thank all reviewers for their helpful comments and criticism on this work. We believe we have addressed the comments and made changes to the methodology and manuscript where possible. We now include supplementary figures and several of the figures in the manuscript have been updated.

Key changes include:

• Analysis and statistics generated for log(AOD) rather than AOD

[Figure]

• Instrument uncertainty included in the estimate

• Regional bias correction of satellite data by AERONET

• Uncertainty in bias correction propagated through analysis

• Marine Aerosol Network (MAN) data included

• Supplementary figures of AERONET and satellite AOD histograms

• Comparison of model AOD with daily AOD from MAN

• Supplementary comparison with deposition flux

The key changes are that the global dust AOD is decreased from 0.033 to 0.030 and the uncertainty increased from 0.006 to 0.011 ($2\sigma$) as a result of considering instrument uncertainty and the uncertainty on the updated AERONET bias correction of the satellite retrievals. The observational estimate is hence closer to the AEROCOM model estimate. We believe that this better corrects for regional biases in the satellite retrievals while representing the inherent uncertainty in using limited in-situ measurements to apply correction factors over large regions. The regional estimates of seasonal dust AOD from the different satellite instruments are generally in closer agreement. The observational estimate is also brought closer to the MERRAero dust AOD; the previous discrepancy was of some concern because MERRAero assimilates MODIS AOD and may be expected to represent the dust AOD better than models without assimilation. The agreement between model and observational estimate improves over the mid-Atlantic, reducing (but not eliminating) the potential for systematically high dust removal in the models. While many of the quoted numbers change as a result of our reanalysis, all other conclusions remain essentially the same.

Please find the reviewer-specific comments and responses (blue italics) listed below.

Kind regards, David Ridley

——————-

[Figure]

Anonymous Referee #5 The manuscript describes a new potential tool for validation of mineral dust in global and regional models, based on a combination of remote sensing data and global climate models. The work is certainly of interest and could provide an additional useful tool to the modeling community. In general the methodology appears sound and the paper is well organized and written. A few minor revision are nevertheless needed in my opinion before the paper could be published.

Thank you for your assessment of the work. We have addressed your concerns below.

————————-

Major comment The construction of the global AOD dataset is the central part of this work. It stems mainly from remote sensing observations, form both satellites and ground-based AERONET stations. I think that too little information is provided regarding data processing (e.g. temporal aggregation) and uncertainties in these types of observations and their relation to dust AOD.

In Section 3.1, we have added more information to the methodology on the process of developing the seasonal AOD from observations and the revised bias correction process using AERONET. We discuss the aerosol properties assumed in MODIS and MISR retrievals in Sections 2.1 and 2.2. In addition, we have incorporated the instrument AOD retrieval uncertainty into the bootstrapping process and acknowledge uncertainties related to these factors in Section 4.4 "Discussion of the remaining uncertainties". From that section:

"Some of the discrepancy between the dust AOD from models and observations is likely born out of simplifications in representing particle morphology and minerology and the resulting impact on the AOD. The models in this study assume a globally fixed refractive index for dust and either spherical or spheroid particle shapes. We do not quantify the uncertainty from mineralogy and morphology here; however, several studies have shown the influence of refractive index and shape upon the derived optical and radiative properties (e.g. Balkanski et al., 2007; Kalashnikova and Sokolik,

2004; Scanza et al., 2015). Scanza et al. (2015) estimate a reduction of approximately 6% on the global dust AOD when accounting for spatially varying mineralogy in the Community Atmosphere Model (CAM-5). Particle morphology and minerology may also present a general bias in AOD retrievals as well as the models. Simplified particle shape modeling during retrieval has been shown to cause underestimation of AOD from space-based retrievals and overestimation from ground-based observations (Kalashnikova and Sokolik, 2002). Similarly, strongly absorbing dust can result in underestimation of the AOD, although improvements in MODIS Collection 6 have been shown to alleviate this (Hsu et al., 2013). The impact on the observational estimate of dust AOD will be dependent upon the specific assumptions made by the MODIS and MISR retrievals, both of which take particle non-sphericity into account but using different methodologies (see Sections 2.1 and 2.2 and references therein). Finally, potential biases exist via erroneous filtering of thick dust plumes during the retrieval (Baddock et al., 2016)."

——————————

Specific comments 2, 8-9: please add a reference here. Added

——————————

2, 14-16: Why PM2.5 in particular? You do not discriminate the size in your product. Good point, thank you. We now just discuss in terms of PM

——————————

2, 24: It would be useful to mention already here what is the general strategy of the work, and why you will use all of the following data from observations or model. Maybe add a table or a brief description in the text, so that the reader can already have a better idea of the role of each type of data in this paper. We have elaborated on the usage of the data products in the introduction to the data description:

"To derive an estimate of dust AOD we make use of AOD retrievals from three satellite instruments as well as surface-based sun photometers to provide a 'ground-truth' for correcting the satellite retrievals. We use in combination with four global aerosol models that provide information on a range of estimates for the non-dust aerosol AOD and the spatial distribution of dust aerosol (see Section 3 for a full description of the methodology)."

—————————-

3, 15: the usage of the angstrom exponent is not clear, please rephrase. Rephrased as follows: "The wavelength-dependence of the AOD, described by the angstrom exponent (Ångström, 1964) between the AOD at 440 and at 870 nm, is used to distinguish AOD dominated by coarse aerosol that is indicated by a lower angstrom exponent than for fine aerosol (e.g. O'Neill et al., 2001; Reid et al., 1999)."

—————————-

4, 24: this sentence is not clear; also the reference is missing from the list. Rephrased and reference added

—————————-

8,20-24: How is your central estimate derived? Is it the mean of the distribution derived from the set of all possible combinations of models and satellite data depicted in Figure 4? Also, please describe more in detail how all the combinations were constructed in the previous section.

We have clarified the methodology in the closing paragraph of the methodology description as follows:

"This process is repeated for all combinations of the 3 satellite instruments, 4 model estimates for non-dust, and 4 model regional-to-global scaling factors; this produces 48 realizations, 16 per satellite instrument, each with an uncertainty estimate. We use the kernel density estimation method (Silverman, 1986) with a Gaussian kernel and standard smoothing to determine a probability density function for the global dust AOD

based on the 48 realizations."

And clarified the description of Figure 3 in Section 4.1:

Figure 3 summarizes our observationally-constrained global dust AOD estimate, averaged over the 2004 – 2008 period, for the combination of all data and for each of the satellite instruments individually.

[Figure]

---

## Author Comment (AC5) · 27 Oct 2016

Response to Reviewers

We would like to thank all reviewers for their helpful comments and criticism on this work. We believe we have addressed the comments and made changes to the methodology and manuscript where possible. We now include supplementary figures and several of the figures in the manuscript have been updated.

Key changes include:

 c Analysis and statistics generated for log(AOD) rather than AOD

• Instrument uncertainty included in the estimate

• Regional bias correction of satellite data by AERONET

• Uncertainty in bias correction propagated through analysis

• Marine Aerosol Network (MAN) data included

• Supplementary figures of AERONET and satellite AOD histograms

• Comparison of model AOD with daily AOD from MAN

• Supplementary comparison with deposition flux

•

The key changes are that the global dust AOD is decreased from 0.033 to 0.030 and the uncertainty increased from 0.006 to 0.011 ($2\sigma$) as a result of considering instrument uncertainty and the uncertainty on the updated AERONET bias correction of the satellite retrievals. The observational estimate is hence closer to the AEROCOM model estimate. We believe that this better corrects for regional biases in the satellite retrievals while representing the inherent uncertainty in using limited in-situ measurements to apply correction factors over large regions. The regional estimates of seasonal dust AOD from the different satellite instruments are generally in closer agreement. The observational estimate is also brought closer to the MERRAero dust AOD; the previous discrepancy was of some concern because MERRAero assimilates MODIS AOD and may be expected to represent the dust AOD better than models without assimilation. The agreement between model and observational estimate improves over the mid-Atlantic, reducing (but not eliminating) the potential for systematically high dust removal in the models. While many of the quoted numbers change as a result of our reanalysis, all other conclusions remain essentially the same.

Please find the reviewer-specific comments and responses listed below.

Kind regards, David Ridley
* * *
Amato Evan This manuscript describes a method of combining satellite and model data in order to estimate the global dust AOD (DAOD). The principal idea here is that models do a good job of simulating non-dust AOD, and satellites do a good job of retrieving the total AOD, so the difference between the two should be a good estimate of DAOD. While I applaud the authors on their creative effort, and the obviously massive amount of time undertaken to complete this work, I find there to be a couple of major issues with the methods that I suspect are contributing to a bias in their global DAOD estimate, and increase the uncertainty. Thus, I am suggesting a major revision.

Major Comments 1. A major assumption of this method is that model DAOD is biased, but that model AOD is not. However, this assumption, at least the part about model AOD not having any systematic bias, isn't justified. The authors suggest that they are accounting for errors related to underestimation of the non-dust AOD by reporting their global DAOD with a 2-sigma uncertainty range (P13, L15). However, if the models systematically underestimate the non-dust AOD, this will induce a high bias in their reported global DAOD, and thus simply increasing the uncertainty range isn't really appropriate. We need to know if there is a bias, particularly because a low bias in modeled non-dust AOD would serve to push the hybrid global DAOD estimate closer to the aerocom mean, and possible closer to the MERRAaero estimate. One could determine if such a bias exists by comparing histograms of AOD for the models and AERONET, over land regions and over-water regions where there is no dust (but there is smoke, anthro. aerosols, and marine aerosols). The difference in those histograms can be used to calculate a bias (which could be corrected) and uncertainty in the models' non-dust AOD. These errors can then be carried through to the final global DAOD calculation.

>A major assumption of this method is that model DAOD is biased, but that model AOD is not. However, >this assumption, at least the part about model AOD not having any systematic bias, isn't justified.

This is certainly a valid concern. We use multiple models to estimate the uncertainty and consider regions where dust aerosol dominates the AOD to minimize the impact of errors in modeled non-dust AOD. For example, the Gulf of Guinea region is not considered explicitly because of the influence of biomass burning. However, as you point out, this may not be sufficient if all models are biased in the same direction. The difficulty is in isolating cases in the dust-influenced regions we consider, but when the dust AOD is not significant. For example, even if we look during wintertime in the Middle East, when dust emissions are low, there is no guarantee that the non-dust AOD will adequately represent the non-dust present during the summertime (looking in other regions, as suggested, is problematic as we may just be observing a local bias that is irrelevant to the dusty locations).

To explore potential biases in modeled non-dust AOD we separate the daily coincident AERONET and model AOD based on whether the model dust AOD contributes >60% or <60% of the total AOD. There is indeed evidence of a bias in CESM and GEOS-Chem. From the added supplementary materials:

"We find that there is a bias between these two cases where CESM and GEOS-Chem both underestimate the AOD relative to AERONET in low dust cases and overestimate the AOD in high dust cases. WRF-Chem and MERRAero show a smaller bias in the opposite direction. Relative to AERONET, the models are biased by -23%, -20%, +3%, +10% (GEOS-Chem, CESM, WRF-Chem and MERRAero, respectively) for the low dust cases, and biased +33%, +12%, +14% and +6% for the high dust cases. The days with low dust AOD in the models are biased low most at AERONET sites in the Thar Desert and Kyzyl Kum, that have limited AERONET data, in the Middle East, and across Africa. This suggests that the non-dust AOD in the models may be biased low on average, which would lead to a high bias in the observational estimate of the dust AOD.

If we re-run the analysis including a regional bias correction factor for the models, we find that the mean estimate of the global dust AOD is reduced to 0.028, a 7% decrease

but still well within our uncertainty estimate. However, with the bias correction applied, the observational estimate of dust AOD in the Mid-Atlantic is unrealistically close to zero in winter and can end up being consistently negative in the Thar desert, suggesting the bias correction is overcompensating. The agreement in seasonal dust AOD between different satellite-model realizations is also worsened, rather than improved. Finally, there is no guarantee that the model dust AOD is an adequate filter to partition the data into low/high dust days, the filter may simply select for seasons when less dust present, which might not tell us much about the non-dust AOD in seasons when dust is present. For this reason, we do not bias-correct the model non-dust AOD for the observational estimate of global dust AOD presented in this work. However, we highlight this potential source of uncertainty in the main text and included a reference to this supplementary text in the summary of explored biases and uncertainty (Table 2)."

In the main text (Section 4.4, pg 15) we add:

"…the non-dust AOD in all models may be systematically biased high or low, which would bias the observational estimate of the dust AOD low or high, respectively. Comparison between modeled and observed AOD at the AERONET sites and MAN ship locations does suggest a low bias in the modeled total AOD in some of the regions considered, although there is no clear systematic bias in the models (see Figures S5 – S9). Comparison of model and AERONET AOD in low and high dust cases (using the model dust AOD to discriminate) suggests that two of the models are biased high and two biased low (Figure S4). Overall, the ensemble of models appears to underestimate the non-dust AOD; correcting this results in a 7% decrease in the global dust AOD estimate (0.028). However, the uncertainties involved in this method are such that we do not include the bias correction in our final estimate (see Supplementary Materials)."
* * *
2. I am also very concerned about use of the models' spatial structure of DAOD (the

horizontal pattern of long-term mean DAOD). In Eqn 2 the authors rely on the spatial structure of modeled DAOD in order to estimate their hybrid global DAOD. The implicit assumption is that while the models' may exhibit biases in the absolute value of DAOD, they do a good job of reproducing the long-term mean spatial structure. However, later on in the paper (P11, Section 4.3) the authors examine the signs of the difference between modeled DAOD and that from their hybrid method in Fig 9 (Africa, N Atl, Gulf of Guinea), suggesting that the models emit too much dust at the source to compensate for the fact that wet and dry deposition is far too strong. So on the one hand you are saying that the spatial structure of model DAOD is good (Eqn 2) and on the other hand it's not (Fig 9). If your hypothesis is correct, that the models emit too much dust because deposition is too strong, then Eqn 2 will introduce a bias into your global DAOD estimate depending on the relative fraction of regions (Fig 1) that are over dust emitting areas and those that are downwind. I think this means that because your regions in Figure 1 are overwhelmingly near or over dust sources, your final global DAOD estimate could be biased low? I'm not entirely sure. . . But the bottom line is that, given this bias in the spatial structure of dust from the models, there is an additional source of uncertainty in the global DAOD estimate, and potentially a bias, related to the distribution of the regions you choose (Fig 1). I'm not exactly sure how you can address this. Maybe add more over-water regions and redo the estimate only using over-water regions, the only using over-land regions, then using both (via Eqn 2)? Or maybe the way to address this potential bias/uncertainty is to recalculate global DAOD using an equal distribution of regions over dust sources and regions downwind of dust sources (also in Eqn 2).

Yes, this is a fair point. The short answer is that there is could be a bias, assuming the excessive removal we infer in the Atlantic is a global issue, but comparison of satellite and model DAOD in remote locations is not possible. We were unable to use the bootstrapping method to determine satellite DAOD over remote regions simply because the DAOD is too low relative to other aerosol AOD and the retrieval sensitivity is too weak (we compared the satellite-retrieved AOD to the MAN network and found that the agreement is much poorer and of limited use over remote locations in the

Southern Ocean and Arctic Ocean).

We compare the models with the MAN observations in remote locations to see if there is an obvious low bias. While correlation is poor in remote regions, there is no clear systematic bias present in the models. However, it is not clear how much of a role dust versus non-dust aerosol plays in this. We refer to previous comparisons of modeled and observed dust surface concentration that show considerable spread in the agreement in remote regions (see Figure 4 of Huneeus et al.; 2011) that limit our ability to discern whether the models used here are unbiased in their representation of the local-to-remote dust distribution. We have added the supplementary figures of comparison of AOD between models and AERONET and MAN, and in the manuscript we have more clearly highlighted this uncertainty in Section 4.4 ("Discussion of the remaining uncertainties"):

"Modeled dust AOD is used as a scaling factor to determine the global dust AOD from the regional observational estimates. We use multiple models to represent the uncertainty, but there may be a systematic bias present, rather than the $\pm6\%$ uncertainty presented (Table 2). If the over-zealous removal of dust in models, highlighted in the mid-Atlantic, is a global phenomenon then the models would predict too much dust in the source regions relative to downwind and yield a low regional-to-global scaling factor. Similarly, dust emissions schemes currently used in the models are unlikely to reproduce emissions where vegetation cover is variable and will not represent dust from agricultural regions (Ginoux et al., 2012). If those emissions are substantial, then it is possible that tuned emissions in models overestimate emissions from large, permanent dust sources to compensate for the lack of agricultural emissions, which could partially explain model bias towards African emissions.

‾‾‾‾‾‾‾‾‾‾‾‾

3. Lastly, I think models report AOD even in the presence of 100% cloud cover. So, in the model, there could be an aerosol layer overlaying stratus clouds, and the model

would save an AOD value. However, in the satellite world, there would be no AOD retrieval. Does this discrepancy induce a bias? Can you examine the model data (I guess you'd need daily or hourly output) to see?

Yes, we have performed this analysis to compare the DAOD with all GEOS-Chem data and with the model data masked where any grid box in a column has >50% cloud cover (based on MERRA reanalysis). We find that the change from masking is small, resulting in a 2% increase in the DAOD. We now mention this explicitly in the main text (pg 11):

"We also calculated GEOS-Chem global dust AOD after masking columns that have >50% cloud cover in any grid box, based on MERRA reanalysis. This causes the global dust AOD to increase by 2%, relative to when no masking is used, indicating that the difference between clear-sky and all-sky dust AOD is small. However, we acknowledge that poor representation of clouds in the reanalysis meteorology or potential satellite misclassification of heavy dust loading as cloud (Darmenov and Sokolik, 2009) could lead to a stronger perceived relationship between dust loading in cloudy and clear sky conditions."

also by masking the model AOD with satellite retrievals. The latter effectively masks for clouds as well as for overpass frequency for better comparison between model and observations. Masking the model DAOD with Aqua and Terra has a negligible effect on the global DAOD (<1%) and masking with MISR increases the global DAOD by 1-2%. This rather surprising finding indicates that the model global DAOD is not significantly different whether or not cloudy locations are included. We have made this more explicit in the text as follows:

"We calculate the modeled global dust AOD with and without masking to match the MODIS and MISR sampling, testing whether sampling affects the derived global dust AOD. We find negligible (<1%) changes in the modelled global dust AOD when sampling to the MODIS instruments and an increase of 1 - 2% when sampling to MISR.

Therefore, we determine that sampling frequency is sufficient to represent the AOD in the regions considered. Furthermore, because the masking effectively removes cloudy regions, the very small change in the modelled global dust AOD indicates that there is no obvious bias in the global dust AOD when including regions within cloudy air masses, relative to clear-sky only"
* * *
Minor Comments 1. P7, L4: Spelling, "main" not "man" Changed
* * *
2. Should alpha have a region superscript in Eqn 1? Previously no, but now that the bias correction is regional it has been added.
* * *
3. P11, L25: Why would a lack of convectively driven dust emissions cause an overestimation of DAOD? Seems like it would be the opposite. The convectively driven dust is strongest in summer, so a lack of that source causes an over-emphasis of winter and spring relative to summer. I think the sentence reflects your intuition.
* * *
4. P7, L26: You write, "In the regions analyzed here the AOD is predominantly driven by dust aerosol, limiting the influence of the model non-dust AOD" but this simply isn't true. Region 1 (N. Atl) also has a big biomass burning contribution in the boreal winter. Regions 8 also has a contribution from anthro. aerosols from N. India during the dry monsoon season. Same for region 10 (from Pakistan and Iran).

Yes, this statement does overemphasize the importance of dust aerosol. We have softened the language as follows:

"In the regions analyzed here dust aerosol plays a key role and often dominates in the spring and summer, limiting the influence of the model non-dust AOD. Exceptions

to this are in South America, South Africa, and Australia, that have a minimal impact on the global dust AOD, and the Gulf of Guinea, where significant biomass burning aerosol is present (we consider results with and without these regions, see Table 1)."
* * *
5. P10: Cloud filtering: Interesting that you are getting such a strong correlation between the two. Misclassification of optically thick dust as cloud may be pretty common, FYI.

Yes, we thought this was interesting. We have added a reference to Darmenov and Sokolik (2009) in there to point out that misclassification of optically thick dust may be occurring.

"We acknowledge that satellite misclassification of heavy dust loading as cloud may occur (Darmenov and Sokolik, 2009) potentially leading to a stronger relationship between dust loading in cloudy and clear sky conditions."

---

## Author Comment (AC6) · 27 Oct 2016

Response to Reviewers

We would like to thank all reviewers for their helpful comments and criticism on this work. We believe we have addressed the comments and made changes to the methodology and manuscript where possible. We now include supplementary figures and several of the figures in the manuscript have been updated.

Key changes include:

• Analysis and statistics generated for log(AOD) rather than AOD

 c Instrument uncertainty included in the estimate

 c Regional bias correction of satellite data by AERONET

 c Uncertainty in bias correction propagated through analysis

 c Marine Aerosol Network (MAN) data included

 c Supplementary figures of AERONET and satellite AOD histograms

 c Comparison of model AOD with daily AOD from MAN

 c Supplementary comparison with deposition flux

The key changes are that the global dust AOD is decreased from 0.033 to 0.030 and the uncertainty increased from 0.006 to 0.011 ($2\sigma$) as a result of considering instrument uncertainty and the uncertainty on the updated AERONET bias correction of the satellite retrievals. The observational estimate is hence closer to the AEROCOM model estimate. We believe that this better corrects for regional biases in the satellite retrievals while representing the inherent uncertainty in using limited in-situ measurements to apply correction factors over large regions. The regional estimates of seasonal dust AOD from the different satellite instruments are generally in closer agreement. The observational estimate is also brought closer to the MERRAero dust AOD; the previous discrepancy was of some concern because MERRAero assimilates MODIS AOD and may be expected to represent the dust AOD better than models without assimilation. The agreement between model and observational estimate improves over the mid-Atlantic, reducing (but not eliminating) the potential for systematically high dust removal in the models. While many of the quoted numbers change as a result of our reanalysis, all other conclusions remain essentially the same.

Please find the reviewer-specific comments and responses listed below.

Kind regards, David Ridley

———————————

Comments from Paul Ginoux

This is a very nice work, which will provide a better constrained mean dust load and optical depth. Still, I wonder about some biases related to satellite data in general, and MODIS Deep Blue in particular. The authors note a lack of bias in MODIS AOD based on the scatter plot of daily values at AERONET sites. However, these sites are characterize by different aerosol environments and surface albedo. Uncertainties related to satellite retrieved AOD between sites will be different. In Figure 3 of Ginoux et al. (Rev. Geophys., 2012), you will notice very different biases between regions. For example Australia is biased high, while Africa is slightly biased low. Although this study was done with Collection 5.1, similar results are obtained with Collection 6, but with much more reduced bias in Australia. My point is that there is very little information we can extract from your Figure 2. A better approach would be to also plot seasonal variation at dusty sites (e.g. Tamanrasset, Birdsville, Solar Village. Dunhuang, etc.).

Thank you for your comments on this work. The simplistic bias correction method that we applied is certainly a source of uncertainty. In response to the review comments we have revised the methodology to apply bias corrections that are specific to each region. We have revised the bias correction of the satellite data to be regional, rather than global and to incorporate Marine Aerosol Network (MAN) daily AOD in relevant regions. We assess an uncertainty to the bias correction by calculating the standard deviation in bias corrections for each year with sufficient (>100) coincident satellite and AERONET retrievals. There are still significant uncertainties resulting in this methodology as the AERONET sites will not represent the entire region, but we believe this goes some way to addressing your concerns and improving the correction applied to each region.

(We did also attempt a bias correction that used the lag-correlation of the AOD from GEOS-Chem to propagate the bias correction to regions surrounding the AERONET site. However, this did not take into account the surface reflectance influence on satellite retrieval bias (which would not share the same lag-correlation). We did not believe there was enough justification for using this approach without also accounting for the

influence of surface reflectance, which was out of the scope of this project).

———————————

I am also concerned about your method of temporal average of observations. If you consider only days with retrievals you will have a high bias, as you discard all days with dust being washout and rainout (low dust). This will be also true for AERONET data. But, it is unclear which method you are using.

Apologies that this was unclear in the paper. We did indeed look at this impact. Firstly, we sample the models to the satellite retrievals and compare the resulting dust AOD with and without sampling. Globally, sampling increased the modeled dust AOD by <1% for MODIS Aqua and Terra and 1-2% when sampling to MISR. Even on a regional basis the sampling bias is less than a few percent. This is obviously only sampling the 'model world' so may not represent reality. If we use the MERRA reanalysis meteorological fields to compare the model dust AOD with and without cloudy regions (masking columns containing grid boxes with >50% cloud cover) we still don't see a large impact (2%), other than in the Gulf Of Guinea where cloud cover is persistent. Intuitively, we would expect there to be a bias between in-cloud and clear-sky dust but this suggests it is more balanced than expected over the timescales considered, probably through many compensating factors.

We have added the following to highlight this potential bias and the work we did to test the effect by masking the model with satellite data and with MERRA cloud cover (pg11):

"We calculate the modeled global dust AOD with and without masking to match the MODIS and MISR sampling, testing whether sampling affects the derived global dust AOD. We find negligible (<1%) changes in the modelled global dust AOD when sampling to the MODIS instruments and an increase of 1 - 2% when sampling to MISR. Therefore, we determine that sampling frequency is sufficient to represent the AOD in the regions considered. Furthermore, because the masking effectively removes cloudy

regions, the very small change in the modelled global dust AOD indicates that there is no obvious bias in the global dust AOD when including regions within cloudy air masses, relative to clear-sky only. We also calculated GEOS-Chem global dust AOD after masking columns that have >50% cloud cover in any grid box, based on MERRA reanalysis. This causes the global dust AOD to increase by 2%, relative to when no masking is used, indicating that the difference between clear-sky and all-sky dust AOD is small. However, we acknowledge that poor representation of clouds in the reanalysis meteorology or potential satellite misclassification of heavy dust loading as cloud (Darmenov and Sokolik, 2009) could lead to a stronger perceived relationship between dust loading in cloudy and clear sky conditions."
* * *
None of the models simulate dust from agricultural regions or with dynamic vegetation. Their contribution is highly uncertain but may affect your results regionally.

This is certainly true. It will not be an issue inside the regions considered (as we do not use model dust AOD here) but it may contribute to errors in model dust in other regions that will affect the global scaling. We have added the following caveat to highlight this uncertainty:

"Finally, dust emissions schemes currently used in the models are unlikely to reproduce emissions where vegetation cover is variable and will not represent dust from agricultural regions (Ginoux et al., 2012). Therefore, it is expected that the tuned emissions in models will overestimate emissions from large, permanent dust sources to compensate and partially explain the bias towards African emissions."
* * *
Finally, you are most likely using MODIS quality flag 3 (QA=3) aerosol products, as advised by Sayer et al. (2013). However, it is not a good choice over dust sources as clearly shown in Figure 1 of Baddock et al. (Geophys. res. Lett., 2015). This choice of

[Figure]

QA=3 may induce a low bias, if you use all days rather than just days with QA=3. On the other hand, if you divide the sum of all valid AOD by the number of days with QA=3, you will again create a high bias. In fact, it may be very high in some areas. Take a look at the factor 10 difference of frequencies between QA=1 and QA=3 in Figure 1 of Baddock et al. (2015). Hopefully this will help improve your results. Paul Ginoux.

Yes, the Baddock et al. work is very interesting. I spoke with Rob Levy about this, but unfortunately isolating QA=1 is not possible with the Level-3 product; that provides Quality-Assurance weighted AOD (giving a weighting of 1.0 to QA=1, 2.0 to QA=2 and 3.0 to QA=3 data) and standard AOD using no weighting but excluding QA=0 data. The merged ocean-DarkTarget-DeepBlue Level-3 product that we use has QA=3 data over land and QA=1-3 data over ocean. However, we have included reference to the Baddock et al. paper to highlight this possibility and to make sure people using Level-2 data in future studies consider the findings of that work. Thank you.

Addition in the MODIS description:

"The merged Level-3 product uses QA=3 data over land and QA=1-3 data over ocean, where higher quality data is given commensurate weighting. Baddock et al. (2016) show that correlation between the frequency of high AOD and dust source location is actually improved when using only QA=1 data. For data to be considered QA>1 the standard deviation in AOD between 1km retrievals must remain below a threshold of 0.18. Therefore, some legitimate dust-influenced retrievals over source may be discarded when using the Level-3 merged product. However, this is a trade off in terms of improving the quality of the retrieval away from source regions."
* * *